# Pleistocene drivers of Northwest African hydroclimate and vegetation

Nicholas A. O'Mara [1,2✉], Charlotte Skonieczny[3], David McGee [4], Gisela Winckler [1,2], Aloys J.-M. Bory[5], Louisa I. Bradtmiller[6], Bruno Malaizé[7] & Pratigya J. Polissar [8✉]

Savanna ecosystems were the landscapes for human evolution and are vital to modern Sub-Saharan African food security, yet the fundamental drivers of climate and ecology in these ecosystems remain unclear. Here we generate plant-wax isotope and dust flux records to explore the mechanistic drivers of the Northwest African monsoon, and to assess ecosystem responses to changes in monsoon rainfall and atmospheric $pCO_2$. We show that monsoon rainfall is controlled by low-latitude insolation gradients and that while increases in precipitation are associated with expansion of grasslands into desert landscapes, changes in $pCO_2$ predominantly drive the $C_3/C_4$ composition of savanna ecosystems.

[1] Lamont-Doherty Earth Observatory, Columbia University, Palisades, NY, USA. [2] Department of Earth and Environmental Sciences, Columbia University, New York, NY, USA. [3] Université Paris-Saclay, CNRS, GEOPS, 91405 Orsay, France. [4] Department of Earth, Atmospheric and Planetary Sciences, Massachusetts Institute of Technology, Cambridge, MA, USA. [5] Université de Lille, CNRS, Université Littoral Cote d'Opale, UMR 8187, LOG, Laboratoire d'Océanologie et de Géosciences, Lille, France. [6] Department of Environmental Studies, Macalester College, St. Paul, MN, USA. [7] UMR CNRS 5805 EPOC, Université Bordeaux I, 33405 Talence, France. [8] Ocean Sciences Department, University of California, Santa Cruz, CA, USA. ✉email: omara@ldeo.columbia.edu; polissar@ucsc.edu

African monsoonal rainfall has played an important role in human migration and evolution[1–6] and supports agriculture and pastoralism that provides food security for at least 80 million people in the African Sahel today[7]. Increasing population[8] and falling crop yields spurred by decreasing precipitation and rising air temperatures[9] over the last several decades has shed light on the vulnerability of this region to climate change. Wide-ranging rainfall projections for this region[10,11], increasing atmospheric $pCO_2$[12], and new evidence for much greater tree cover than previously thought[13] all underscore the need for improved understanding of the mechanistic drivers of this region's hydroclimate and ecosystem composition.

Several decades of work on African margin marine sediment archives of eolian dust flux demonstrate marked shifts in periodicity, from precessional (19- and 23-kyr) dust cycles during the Pliocene toward a stronger obliquity (41-kyr) component in the early Pleistocene, and ultimately to a 100-kyr beat following the Mid-Pleistocene Transition. These shifts are consistent with an increasingly strong high-latitude influence on the Northwest African monsoon, as Northern Hemisphere ice sheets expanded[1,2,14,15]. Newer biomarker proxy reconstructions of Northwest African paleohydrology also show Pliocene variability dominated by precession and obliquity[16,17], when global ice volume was paced by obliquity. However, during the Middle to Late Pleistocene, when high-latitude climate is largely paced by the 100-kyr cycle, biomarker paleohydrologic records and dust flux records calculated using new constant flux proxy-normalization techniques from this region continue to be dominated by precession and obliquity[16–21]. Such cycles can be driven by low-latitude insolation, latitudinal insolation gradients, or ice sheet-driven atmospheric teleconnections which climate models suggest could shape subtropical monsoon rainfall in Northwest Africa[22–27]. Our understanding of these processes remains limited because only one of these late Pleistocene records extends beyond two glacial cycles, making it difficult to test the degree to which high-latitude vs. low-latitude processes drive monsoon rainfall dynamics throughout the late Pleistocene.

While rainfall regimes largely shape the distribution and ecological makeup of African savanna ecosystems today and in the past, atmospheric $CO_2$ levels can also influence plant growth and ecosystem composition[28–30]. Modern rainfall and vegetation relationships are often applied to interpret paleorecords of vegetation change as indicating changes in past hydrology[16,31–33]. However, higher atmospheric $CO_2$ levels confer competitive advantages to $C_3$ photosynthesizing plants which cannot be discerned from modern spatial relationships[28,29]. Observations of savanna woody cover over the last century[34–36], $CO_2$ fertilization experiments[37], and modern vs proxy and model estimates of last glacial maximum (LGM) vegetation[38,39] all show shifts toward increasingly woody savannas ($C_3$) during intervals of higher atmospheric $pCO_2$, independent of increasing rainfall. These results call into question the interpretation of vegetation change in Northwest Africa as a paleo-aridity indicator. Furthermore, reliable estimates of the future composition of West African savannas require improved understanding of the relative controls of rainfall and $pCO_2$ on vegetation in this region.

Here we examine the combined effects of $pCO_2$ and monsoon rainfall on savanna ecosystems and the underlying controls of high- vs low-latitude forcing of monsoon rainfall in Northwest Africa during the Middle to Late Pleistocene. We sampled marine sediment core MD03-2705 (Fig. 1a) at orbital resolution over marine isotope stages (MIS) 13 to 10 (~520–360 ka), a period of dramatic changes in orbital eccentricity and the largest changes in global ice volume and atmospheric $pCO_2$ over the last 1 million years. We reconstruct monsoon intensity using both plant wax-derived $\delta D_{precip}$ and dust fluxes normalized to the extraterrestrial

$^3$He ($^3He_{ET}$) constant flux proxy, and assess the relative controls of monsoon rainfall and $pCO_2$ on savanna ecosystem structure using plant-wax $\delta^{13}C$ that tracks $C_3/C_4$ vegetation abundance (related to tree/grass cover in African savannas). We find that monsoon rainfall variability and dust emissions are driven primarily by low-latitude insolation gradients, and that while precipitation controls the northward expansion of grasslands into the Sahara Desert, it plays a relatively minor role compared to $pCO_2$ in controlling the composition of savanna vegetation.

## Results and discussion

**$\delta D_{precip}$ reveals dynamics of Northwest African monsoon insolation forcing.** Changes in monsoon rainfall inferred from $\delta D_{precip}$ values exhibit no marked differences between glacial and interglacial intervals and rather appear more similar in pacing to changes in local summer insolation (23.5°N, June 21) (Fig. 1b). The shared variance between local insolation and $\delta D_{precip}$ declines over time from MIS 13 to 10 as orbital eccentricity wanes, reducing the amplitude of the precession forcing, while the amplitude of the monsoon response does not change. Wavelet analysis of the $\delta D_{precip}$ record shows the later portion of the record is dominated by obliquity pacing (Fig. 2a), while local insolation is forced exclusively by precession (Fig. 2c). The disparity between the amplitudes and the lack of strong obliquity-paced variability in the direct insolation forcing means an additional forcing mechanism is required in order to explain the variability in monsoon rainfall.

Multiple modeling studies have observed strong monsoon responses to changes in the obliquity of the Earth's orbit[22–25]. Obliquity pacing of monsoons has been hypothesized to arise via indirect control by the impact of ice sheet expansion on ventilation of cool and dry midlatitude air masses into subtropical monsoon regions[1,14,26,40,41], while other recent work on Northwest African monsoonal rainfall has shown that insolation gradients are a potential driver of obliquity-paced variability during the last glacial cycle[42] and during the Pliocene[16]. Our $\delta D_{precip}$ record does not bear any resemblance to the high latitude temperature and ice volume evolution. The amplitude evolution of the obliquity component of the $\delta D_{precip}$ record does not match the obliquity variability of global ice volume, and the $\delta D_{precip}$ obliquity signal leads the ice volume obliquity signal by ~6 kyrs, precluding ice sheets as the main driver of the observed monsoon response (Fig. S7). Therefore, obliquity pacing in our record must arise from a direct response to insolation gradients, or due to nonlinear responses to Northern Hemisphere insolation variability.

The two proposed mechanisms by which insolation gradients may force monsoon rainfall are both the result of changing heat and moisture transport from obliquity-driven changes in latitudinal insolation distributions. The first is the summer subtropical to high latitude insolation gradient, which is diminished during times of increased axial tilt resulting in decreased northward moisture transport out of the subtropics. This mechanism was originally proposed to explain obliquity-paced variability in Northern Hemisphere ice sheets both before[43] and after[44] the Mid-Pleistocene Transition. The second is the summer inter-hemispheric insolation gradient which links increases in monsoon rainfall during times of high obliquity to enhanced moisture transport from the South Atlantic to the Northwest African monsoon region via strengthened southerly winds associated with an intensified winter hemisphere Hadley cell[22].

Our $\delta D_{precip}$ record is well correlated with the summer inter-hemispheric insolation gradient (23.5°N–23.5°S, June 21, $r = 0.75$, $p < 0.001$) (Fig. 1b) but has only a weak relationship with the

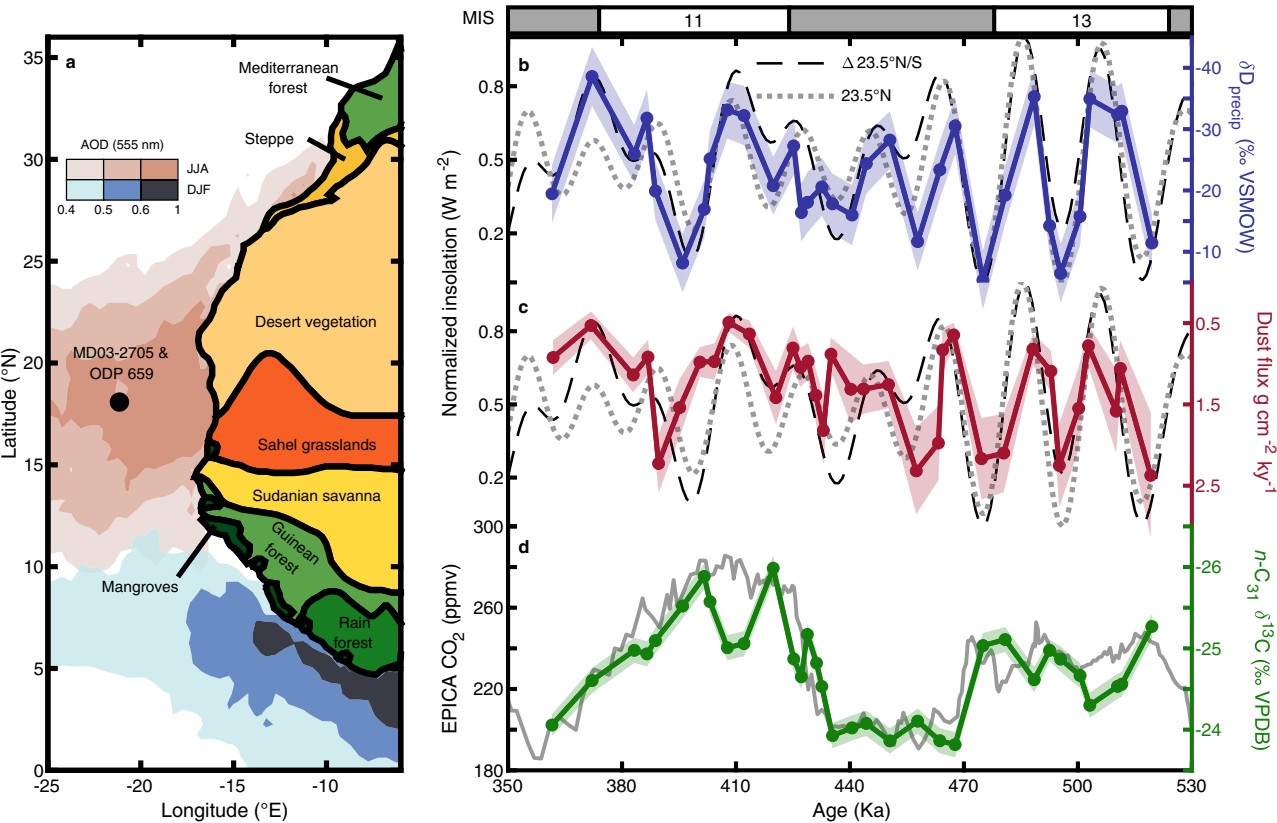

**Fig. 1 Regional context and new records of monsoon rainfall, dust, and vegetation. a** Map of Northwest Africa with biome distributions on land ref. [54], the aerosol optical depth (AOD, 555 nm) (summer: JJA and winter: DJF, 2000–2017) over the ocean (Giovanni, NASA EarthData) showing the location of the modern summer and winter Saharan dust plumes (colorbar, ref. [127]), the core location for MD03-2705 and ODP 659 is labeled with a black circle **b** $\delta D_{precip}$ with 1σ uncertainty plotted with June 21 23.5°N insolation (dotted line, $r = -0.77$) and the summer inter-hemispheric insolation gradient June 21 23.5°N–23.5°S (dashed line, $r = -0.76$), insolation was determined using ref. [126] and normalized to values between 0 and 1. **c** $^3\text{He}_{ET}$-normalized dust flux plotted with insolation curves as in **a** (23.5°N, $r = -0.46$; Δ23.5°N/S, $r = -0.50$). **d** Plant-wax $C_{31}$ $n$-alkane $\delta^{13}C$ with 1σ uncertainty plotted with EPICA ice core $CO_2$ concentration ref. [128] ($r = -0.81$). $\delta D_{precip}$ and $^3\text{He}_{ET}$-normalized dust flux are also well correlated ($r = 0.68$). All correlation coefficients are significant at a $p$ value of <0.01. Marine isotope stages (MIS) are indicated with interglacials periods shown in white and glacial periods in gray.

summer subtropical to high latitude insolation gradient (25°N–65°N, June 21, $r = 0.31$, $p = 0.03$. Wavelet analysis further shows that the $\delta D_{precip}$ variability in the frequency domain is similar to the summer subtropical to high latitude insolation gradient at obliquity periods (41 kyrs) but only the summer inter-hemispheric insolation gradient can explain both the precession- and obliquity-related variability in the $\delta D_{precip}$ record (Fig. 2d, e).

Previous work has attributed this dual obliquity/precession-paced variability in the northwest African monsoon to a hybrid response where changes in local insolation drive precession-paced variability, and the summer subtropical to high latitude insolation gradient drives obliquity-paced variability[16]. Here, we find that the phasing and the scaling of the $\delta D_{precip}$ record are tightly coupled with the summer inter-hemispheric insolation gradient (Fig. 1b). It is difficult to disentangle from these data alone if increased cross-equatorial moisture transport or decreased low to high latitude moisture transport controls the obliquity response of the Northwest African monsoon. However, model simulations of increased obliquity do not show decreases in rainfall in the Northern Hemisphere extra-tropics[22] that would result if this mechanism drove the obliquity variability. In fact, simulations of Mediterranean rainfall[45] show in-phase responses to obliquity forcing with Northwest African monsoon rainfall indicating reduced poleward moisture transport resulting from less steep low to high latitude insolation gradients is not likely the dominant control of obliquity-paced variability of Northwest African monsoon rainfall. On the other hand, the summer inter-

hemispheric insolation gradient can fully explain the monsoon response to low latitude insolation variability. The response of the northwest African monsoon can be conceptualized as a combined response to local insolation driving an increased land-sea temperature gradient and thus increased moisture transport to the monsoon region from the equatorial Atlantic, paced by precession, with a simultaneous additional response to obliquity which drives enhanced cross-equatorial moisture transport from the South Atlantic into the monsoon region. We cannot rule out non-linear responses to small obliquity-paced changes in local insolation. However, due to the agreement between our $\delta D_{precip}$ record of monsoon rainfall coupled with dynamical support for this mechanism from climate model simulations[22,23], we conclude that tropical moisture transport driven by direct thermal responses to low latitude insolation forcing coupled with summer inter-hemispheric insolation gradient driven moisture transport—not high latitude processes—is the main driver of monsoon rainfall.

**Dust flux tracks $\delta D_{precip}$.** The dust flux to our core site shows a remarkable resemblance to our $\delta D_{precip}$ record ($r = 0.70$, $p < 0.001$), tracking changes in the summer inter-hemispheric insolation gradient but not following glacial-interglacial cycles (Fig. 1b, c). Dust fluxes to marine sediments are sensitive to aridity, source area expansion, and/or wind intensity[46]. Contraction of vegetated areas and reduced soil moisture during weak

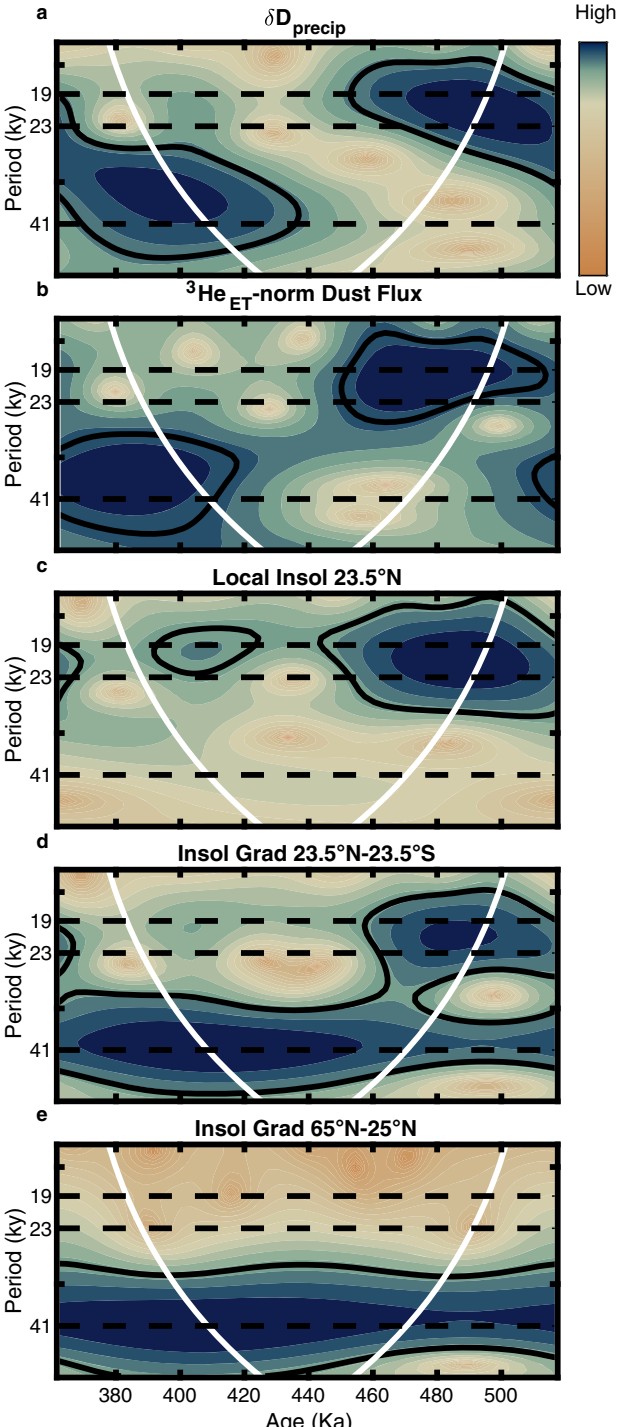

**Fig. 2 δD$_{precip}$, dust flux, and insolation wavelet analysis.** Wavelet scalograms depicting power at different periods through time for **a** δD$_{precip}$ record, **b** $^3$He$_{ET}$-normalized dust flux, **c** local summer insolation, 23.5°N June 21, **d** summer inter-hemispheric insolation gradient 23.5°N–23.5°S June 21, **e** low to high-latitude summer insolation gradient, 25°N–65°N June 21. Continuous wavelet transforms were calculated following ref. [122] with all insolation time series sampled at wax isotope sample time points then interpolated to 3 kyr intervals. In all panels, the white parabola indicates the cone of influence. Black solid lines encompass intervals with power that is significant at the 95% significance level. Only the cross-equatorial insolation gradient can explain both the precessional and obliquity-driven components of the δD$_{precip}$ and the $^3$He$_{ET}$-normalized dust records.

monsoon intervals (see below) expands the total area of deflatable sediment, increasing the potential for higher dust emission. At the same time, stronger winds during dry intervals[47] likely also contribute to higher dust fluxes when δD$_{precip}$ indicates decreased rainfall. The strong link between δD$_{precip}$ and dust flux found here (Figs. 1b, c and 2a, b) nevertheless suggests these two proxy records are both faithfully recording monsoon variability.

Previous dust flux records over the Plio-Pleistocene based on age model-derived mass accumulation rates (MARs) show a strong global ice volume signature suggesting high-latitude forcing of Northwest African monsoon rainfall[1,14]. However, this approach assumes sedimentation rates are uniform between age model tie points which implicitly does not account for variability in syndepositional processes including sediment redistribution (focusing/winnowing). In contrast, constant flux proxy-normalization techniques ($^{230}$Th$_{XS}$ and $^3$He$_{ET}$) calculate instantaneous fluxes which correct for sediment redistribution. A new $^{230}$Th$_{xs}$-normalized dust record from the same core (MD03-2705) found increased spectral power for obliquity and precession compared to the age model-derived MAR technique[20]. This suggests that systematic differences in sediment redistribution between glacial and interglacial intervals may have introduced unknown biases into previous age model-derived dust flux estimates. Unlike previous work which has attributed dust flux variability in this region to high latitude insolation variability, we find both the $^3$He$_{ET}$-normalized dust flux reconstruction from this study and the $^{230}$Th$_{xs}$-normalized dust flux record of ref. [20] (240–0 ka) are consistent with changes in monsoon rainfall and aridity forced by the summer inter-hemispheric insolation gradient (Figs. 3c, d and S6a, d).

**C$_3$/C$_4$ vegetation dynamics reflect combined influence of pCO$_2$ and rainfall.** On the African continent, grassland and savanna ecosystems made up at least in part by C$_4$ grassy vegetation exist within a mean annual rainfall range of ~250–1750 mm/yr, where woody (C$_3$) xeric shrub vegetation dominates below this range and closed-canopy forests above it[30,48]. Within these rainfall limits, the varying physiologies and growth strategies of C$_3$ trees and C$_4$ grasses affect their relative abundance in response to environmental conditions: C$_4$ grasses outcompete C$_3$ trees when growing seasons are warmer[49,50], rainfall is lower or more seasonal[51], when atmospheric CO$_2$ is lower[29,49], or when disturbance by fire or herbivory is high[52]. Here we explore the role of changing rainfall amount and atmospheric pCO$_2$ as potential drivers of landscape-scale C$_3$/C$_4$ balance.

As expected, our record shows higher δ$^{13}$C values––indicating more grassy (C$_4$) vegetation––during low pCO$_2$ glacial intervals. While the extent of global glaciation, mostly controlled by Northern Hemisphere ice volume, is highly correlated with atmospheric CO$_2$ levels, during glacial inceptions CO$_2$ falls rapidly, in line with global temperature, while ice volume increases more gradually[53]. Our new δ$^{13}$C record tracks changes in CO$_2$, not ice volume (Fig. S8), indicating that CO$_2$ and not the extent of glaciation is the primary driver of the observed glacial-interglacial variability in ecosystem C$_3$/C$_4$ balance. There is also a weak correlation indicating more grassy C$_4$ vegetation (more positive δ$^{13}$C) during high monsoon rainfall intervals (more negative δD$_{precip}$) (Figs. 1d, S9). This relationship is counter-intuitive, but has been previously described by ref. [42] as resulting from expanded grasslands into previously sparsely vegetated desert landscapes along the Southern edge of the Sahara during intervals of enhanced monsoon rainfall. During periods of higher monsoon rainfall, such as the African Humid Period[18], both pollen records[54,55] and models[56] show C$_4$ grasslands expand into

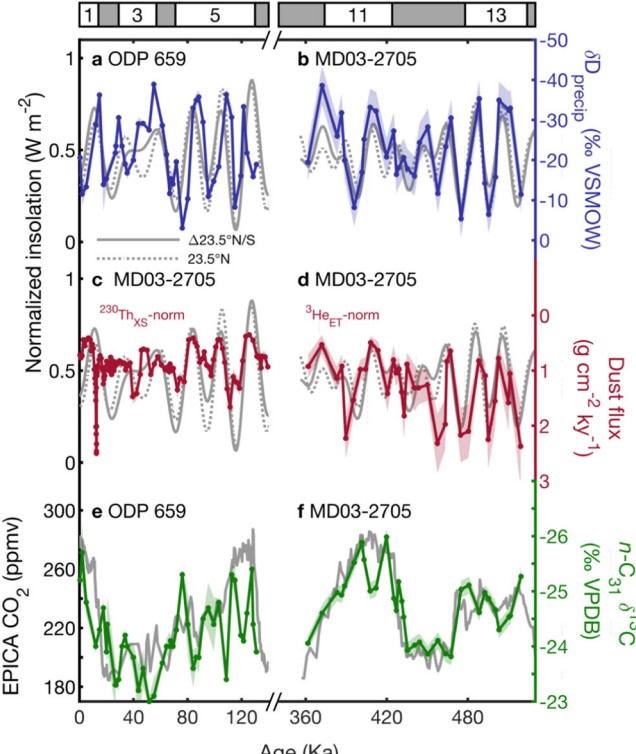

**Fig. 3 Comparison between MIS 13-10 and the last glacial cycle. a** $\delta D_{precip}$ with 1σ uncertainty (ref. [42], blue line and shading) plotted with the normalized summer inter-hemispheric insolation gradient June 21 23.5°N–23.5°S insolation (solid gray line) and the normalized local insolation June 21 23.5°N (dashed gray line), **b** $\delta D_{precip}$ (this study) plotted as in **a**, **c** $^{230}Th_{XS}$-normalized dust flux with 1σ uncertainty (ref. [20], red line and shading) plotted with the summer inter-hemispheric insolation gradient June 21 23.5°N–23.5°S insolation (gray line), **d** $^{3}He_{ET}$-normalized dust flux (this study) plotted as in **c**, **e** Plant-wax $n$-$C_{31}$ $\delta^{13}C$ with 1σ uncertainty (ref. [42], green line and shading) plotted with EPICA ice core pCO₂ (ref. [128], gray line), **f** $\delta D_{precip}$ (this study) plotted as in **c**. Note that MIS 6-1 and 13-10 dust fluxes use different methods that may have additional uncertainties when comparing them (see "methods" and Figs. S10, 11 for details).

from just offshore of the Congo Rainforest shows enhanced $C_3$ vegetation in the LGM compared to the Late Holocene. The physiological advantages that benefit $C_4$ vegetation during times of low atmospheric pCO₂ are likely outweighed in well-established rainforest ecosystems where out-shading by closed-canopy trees prevents grasses from proliferating despite their photosynthetic advantages[63]. Stronger trade winds during the LGM[47,64,65] may have increased plant-wax transport from the Congo Rainforest to the equatorial Atlantic during boreal summer causing a shift toward more negative plant-wax $\delta^{13}C$ values at this equatorial site unrelated to changes ecosystem structure. Nevertheless, these broad patterns illustrate the differential effects of pCO₂ and monsoon rainfall on grassy savanna ecosystems. Enhanced monsoon rainfall expands the northward edge of rain-limited savanna grasslands, while higher pCO₂ favors woody $C_3$ vegetation growth in existing savannas. These results also corroborate similar conclusions drawn in an ecological modeling study from Southwest African savannas[39]. These authors showed that the higher reconstructed abundances of $C_4$ plants during the LGM in Southwest African savannas compared to the late-Holocene[57] required plant physiological responses to CO₂ in addition to changes in monsoon rainfall and temperature. Our new data show that this CO₂-induced physiological driver of $C_3$/$C_4$ balance also holds true in Northwest Africa for at least the last ~500 ka, lending more confidence to the importance of a pCO₂ control on savanna woody cover on the African continent.

We observe different magnitudes in the response of $C_3$/$C_4$ balance to monsoon rainfall and pCO₂ during MIS 13–10 (this study) compared to MIS 5–present at site ODP 659 (adjacent to our site)[42]. During both intervals, pCO₂-driven changes are of similar magnitude. However, despite similar variability in monsoon rainfall ($\delta D_{precip}$) (Fig. 3a, b), the MIS 5–present (120–0 ka) vegetation ($\delta^{13}C$) from core site ODP 659 appears more sensitive to changes in the monsoon than during MIS 13–10 (Fig. 3c, d, ED9). This can be explained by more northerly-shifted ecosystem boundaries during MIS 13–10, which is supported by pollen reconstructions[55]. During MIS 13-10, closer proximity of the core sites to savanna ecosystems where woody ($C_3$) vegetation growth is restricted under lower atmospheric pCO₂ results in a greater impact of pCO₂ compared to rainfall on the regional $C_3$/$C_4$ balance.

**Implications for the future of Northwest African ecosystems.** The combined records of $\delta D_{precip}$ and plant-wax $\delta^{13}C$ presented here show that since at least ~500 ka the physiological control of atmospheric pCO₂ on photosynthesis dominantly controls the woody/grassy balance in existing savannas, while the largest effect of increased monsoon rainfall on Northwest African vegetation is to expand $C_4$ grasslands into the Sahara Desert. This is consistent with recent results suggesting pCO₂ has likely played an important ecological role in the vegetative evolution of Africa during the LGM-Holocene[38,39] and over the Neogene[66]. The new insights from our record suggest that since pre-industrial times, rising atmospheric CO₂ concentrations have likely been a strong forcing on woody cover in African savanna ecosystems, corroborating observational studies over the past several decades which attribute current increasing trends in African savanna woody cover with rising CO₂ levels[34–36]. However, controlled CO₂ release experiments reveal that the effect of CO₂ fertilization on woody cover is non-linear and likely saturates at values higher than ~400–500 ppm[37], which makes the future ecological role of atmospheric CO₂ in savanna ecosystems uncertain. Nevertheless, persistently high concentrations of CO₂ (>400 ppm) in the atmosphere will impose a continued ecological pressure favoring

previously unvegetated areas, increasing the overall area covered by $C_4$ plants.

The relative expression of CO₂-driven changes in savanna $C_3$/$C_4$ makeup and monsoon rainfall-driven expansion of $C_4$ grasslands into unvegetated areas recorded by plant-wax $\delta^{13}C$ will depend upon where a core site is located in relation to existing ecosystem boundaries. Expanding upon previous work[39,57], we have compiled plant-wax $\delta^{13}C$ data from core top and downcore records along the West African margin to assess the spatial relationship between the impacts of pCO₂ and monsoon rainfall on landscape $C_3$/$C_4$ balance (Fig. 4). In comparison to the drier late Holocene (2–0 ka), sediments from the wetter middle Holocene (8–6 ka) have more positive plant-wax $\delta^{13}C$ values poleward of ~15°N, consistent with $C_4$ grass expansion into the Sahara Desert. Mid-Holocene $C_3$ woody contributions increase at equatorial latitudes consistent with poleward movement of the forest/savanna boundary (Fig. 4a, b). During the LGM (23–19 ka), when atmospheric CO₂ concentrations were ~90 ppm lower but rainfall as estimated by plant-wax $\delta D$[42], runoff from the Niger River[58], and paleolake levels[59–62] were broadly similar to modern (see methods for more details), at nearly all latitudes there is a higher proportion of $C_4$ plants compared to the late Holocene (Fig. 4c, d). One sediment core

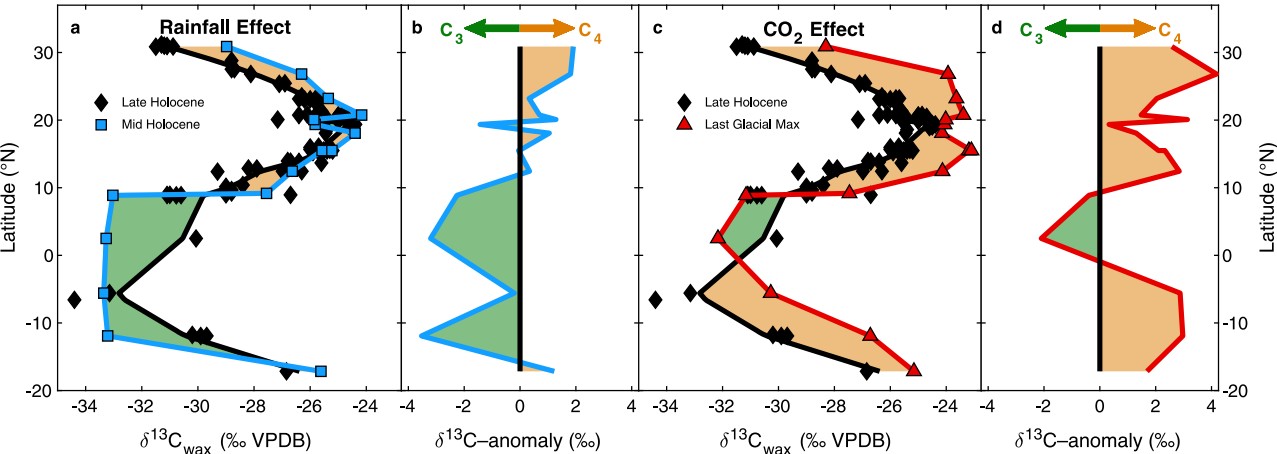

**Fig. 4 Late Holocene, Middle Holocene and LGM patterns in latitudinal C₃ and C₄ vegetation distributions.** Plant-wax δ¹³C along the West African margin refs. [6,18,42,57,94,129–131] separated into late Holocene (2–0 ka, black diamonds), middle Holocene (8–6 ka, blue squares), and Last Glacial Maximum (23–19 ka, red triangles). Each transect is fit using a locally weighted scatter-plot smoothing (LOESS) function, ("smooth" function, MATLAB v2020a, see "methods" for details). The relative contribution of C₄ v. C₃ vegetation between intervals is shown in orange (greater C₄) and green (greater C₃) shading. **a** The effect of increased rainfall during the humid middle Holocene. **b** Middle Holocene – Late Holocene δ¹³C$_{wax}$ anomalies calculated for each individual core **c** The effect of decreased atmospheric CO₂ during the LGM. **d** LGM – Late Holocene δ¹³C$_{wax}$ anomalies calculated as in **b**.

tree over grass growth. The abatement of woody encroachment on savannas will thus be a more difficult task, posing distinct challenges to agropastoralist communities that rely on savanna ecosystems for livestock grazing.

## Methods

**Site, samples, and source region.** The IMAGES Calypso core MD03-2705 was taken in the eastern equatorial Atlantic off the Mauritian Coast (18°05′N, 21°09′W) at a water depth of 3085 m below sea level[67]. The core site is located at the apex of a 300 m seamount between the Cape Verde Islands and the Northwest African margin and is under the direct influence of the present-day Saharan dust supplies[68]. Regarding this particular bathymetric and geographical setting, previous work has shown that non-carbonate siliciclastic input[69], as well as plant waxes, are primarily delivered to core sites in this region by easterly winds from the adjacent Western African continent with dust sources constrained to the western and central Sahara[69] and plant waxes constrained to the African continent primarily from South of the Sahara (see "Interpretation of plant-wax isotopes" section below for details) (Fig. 1). The core was sampled to target precession cycles and measurements of plant-wax compound-specific carbon and hydrogen isotopes and n-alkane distributions (n = 33) as well as ³He$_{ET}$-normalized dust fluxes (n = 37) which were made at ~10–20 cm intervals corresponding to a temporal resolution of ~3 kyrs.

**Age model.** The age model for core MD03-2705 was determined via peak-to-peak matching of the benthic foram δ¹⁸O record from this core[70] to the LR04 global benthic stack[71] (Fig. S1, Supplementary Data 1). Three paleomagnetic reversal ages corresponding to the lower and upper Jaramillo and the Matuyama/Bruhnes reversals provide additional age control in the lower portion of the core[70]. The oxygen isotope stratigraphy produced a good match the LR04 global benthic stack (r² = 0.85; Fig. S1) with little variation in linear sedimentation rate. One spike in sedimentation rate was observed between 19.90–20.53 meter below seafloor (mbsf), however, this interval corresponded to a dark muddy and homogenous layer in the core which likely corresponds to an event of rapid (or instantaneous) deposition.

**Plant-wax n-alkane biomarker extraction and quantification.** Wet sediment samples were freeze-dried and lipids were extracted from ~20 g dry and crushed sediment aliquots using a Dionex 200 Accelerated Solvent Extractor (ASE) using DCM:methanol (9:1 v/v). Total lipid extracts (TLEs) were spiked with an internal standard mixture of known concentrations (5α-androstane, 1-1′ binaphtyl, stearyl stearate, 11-eicosenol, 11- eicosenoic acid) and dried gently under N₂ gas before column chromatography. TLEs were loaded onto silica gel columns (~0.5 g, Sigma–Aldrich 70-230 mesh, 60 A; DCM, MeOH extracted, activated 2 h at 200 °C) and separated in aliphatic, ketone, and polar fractions using hexane (4 ml), DCM (4 ml), and methanol (4 ml) as eluents, respectively. Long-chain n-alkanes in the aliphatic fractions were quantified by gas chromatography mass-spectrometry on an Agilent 7890A GC coupled to an Agilent 5975C MS with a DB-5 silica capillary column (30 m × 25 mm inner diameter, 0.25 μm film thickness) using a full scan method with an electron impact ionization of 70-eV. Compounds were identified by characteristic ion fragments and by comparing retention times to an external n-

alkane standard and integrated using Chemstation (Agilent) software. Concentrations of n-alkanes were determined by comparison to an internal 5a-androstane recovery standard using a response factor for the ion m/z 57 calculated from an external standard mixture of C₈–C₄₀ n-alkanes run every six samples. Concentrations of long-chain (C₂₁–C₃₅) n-alkanes displayed high odd-over-even preference (5.80–7.54) with a maximum peak at C₃₁ consistent with a terrestrial plant biomarker source. Concentrations were further used to determine dilution volumes for each sample before compound-specific carbon and hydrogen isotope analysis. All concentrations can be found in Supplementary Data 2.

**Compound specific n-alkane δD and δ¹³C.** Compound specific carbon (δ¹³C) and hydrogen (δD) isotope measurements on n-alkanes in the aliphatic organic fraction were performed using a Thermo Delta V advantage isotope ratio mass spectrometer coupled to a Thermo Trace GC Ultra and Isolink through a ConFlo IV. Each sample in 1–4 μl of hexane was injected into a PTV injector with 2 mm i.d. silicosteel liner packed with glass wool. The inlet was run in splitless mode at 60 °C during the injection then ramped to 320 °C and held for 1.5 min during the transfer phase. Separations of individual n-alkanes were done on an HP-5MS column (30 m length, 0.25 mm i.d., 0.25 μm phase thickness) with a constant helium flow of 1.0 ml/min. The GC oven settings were as follows: 60 °C hold of 1.5 min ramped at 15 °C/min to 150 °C and then at 4 °C/min to 320 and held for 10 min.

Carbon isotopic compositions of n-alkanes were measured via combustion to CO₂ after chromatographic separation via a custom-made combustion reactor consisting of one strand each of nickel, copper, and platinum wires inside a 0.5 mm i.d. fused alumina tube held at 1000 °C. The reactor was initially oxidized with pure O₂ gas and a trickle of 1% O₂ in helium was introduced inline prior to the combustion reactor to ensure continued oxidation and complete combustion of compounds. Simultaneous measurements of n-alkanes of known carbon isotopic composition (purchased from Arndt Schimmelmann, University of Indiana) were performed between every six samples to determine individual compound δ¹³C values on the VPDB scale.

Hydrogen isotopic compositions of n-alkanes were measured via pyrolysis to H₂ via a rector purchased from Thermo-Fisher Scientific consisting of an 0.5 mm i.d. open alumina tube connected in the GC oven to the GC column through a silicosteel capillary. The reactor was conditioned with two injections of 1 μl hexane. Simultaneous measurements of n-alkanes of known hydrogen isotopic composition (purchased from Arndt Schimmelmann, University of Indiana) were performed between every six samples to determine individual compound δD values on the VSMOW scale.

Replicates of every sample for both hydrogen and carbon analysis were run to assess reproducibility. Conversion of δ¹³C and δD to the VPDB and VSMOW scales respectively and their uncertainties were calculated following ref. [72].

**Interpretation of plant-wax isotopes.** Change in the δ¹³C of atmospheric CO₂ can bias estimates of plant-wax δ¹³C across glacial transitions. A recent compilation of δ¹³C$_{CO_2}$ over the last 150 kyrs encompassing the last glacial cycle found variability of only ~0.5‰ between glacial and interglacial conditions, with more enriched δ¹³C$_{CO_2}$ during glacials (LGM and MIS 4) and more depleted δ¹³C$_{CO_2}$ during interglacials (Holocene, MIS 3, MIS 5)[73]. Direct measurements of δ¹³C$_{CO_2}$ variability across MIS 13–10 have not been made, however, seem unlikely to have

varied much more than over the last glacial cycle given the similarities in pCO$_2$ change across this interval. Here, plant-wax δ$^{13}$C values are not corrected for changes in δ$^{13}$C$_{CO_2}$ which would result in more negative values during interglacials and more positive values during glacials, and thus the record we present here is a conservative estimate of changes in vegetation during this interval.

The plant-wax δD record of the C$_{31}$ n-alkane was corrected for changes in global ice volume across MIS 13–10 following ref. [74] by scaling the LR04 benthic δ$^{18}$O stack[71] of the last glacial maximum to 1‰ ref. [75]. The scaled δ$^{18}$O values, representing changes in seawater oxygen isotopes (δ$^{18}$O$_{sw}$), were linearly interpolated to the ages of the plant-wax samples and converted to δD via the global meteoric water line (8:1 ratio between δD and δ$^{18}$O in modern precipitation) and used to compute ΔδD$_{sw}$. The ice volume corrected plant-wax δD record (δD$_{wax,IVC}$) was then calculated with the following equation:

$$\delta D_{wax,IVC} = (\delta D_{wax,IVC} + 1)/(\delta D_{sw} + 1) - 1 \qquad (1)$$

The carbon isotope composition (δ$^{13}$C) of plant waxes deposited in marine sediments in the eastern Atlantic is sensitive to plant physiology in Western Africa, namely the photosynthetic pathway used by the wax producer for fixing organic carbon (e.g., C$_3$ v. C$_4$)[33,76]. Here, we interpret changes in the plant-wax δ$^{13}$C as primarily representative of changes in the percent landscape cover of C$_4$ grass v. C$_3$ trees using the δ$^{13}$C of the C$_{31}$ n-alkane[77], but changes in local hydrology, temperature, and pCO$_2$ can all potentially impact the δ$^{13}$C of plant waxes.

The provenance of the plant waxes at our study site has important implications for the interpretations of our paleoecology and paleohydrology reconstructions. The range of the plant-wax (C$_{31}$ n-alkane) values presented here (−25.98 to −23.82‰) implies a mixed C$_3$/C$_4$ vegetation source to our core site with grassy C$_4$ vegetation making up ~56–72% of the total (see below for calculations). Today, outside of the Sahara Desert where vegetation is sparse two potential plant-wax source areas, or "wax-sheds" are possible: the Sahel grasslands, savannas, and deciduous forests to the south (savanna source) or the steppe and Mediterranean forests to the north (Med source). While the only substantial source of C$_4$ plant material is from the savanna source to the south, C$_3$ plants occur in both the Med and savanna source regions[78]. During African Humid Periods when the Northwest African monsoon is amplified, it is well documented that grassland, savanna, and deciduous forest ecosystems south of the Sahara expand northward, with the most dramatic change occurring as grasslands expand into the Sahara Desert increasing the relative contribution of C$_4$ plant-waxes to our core site[54–56,79–82]. Conversely, during weak monsoon intervals vegetation zones are shifted south with the most significant change being the contraction of grasslands out of the Sahara Desert region.

During the most recent weak monsoon interval, there is evidence for increased Med source pollen as a proportion of the total pollen farther south along the west African margin. This has been interpreted as evidence for strengthened trade winds and greater transport of Med source pollen to sediments in the range of 16–22°N. The same data have been used to argue for a greater contribution of Med source plant-waxes offshore Northwest Africa during weak monsoon intervals[42,54]. Within this framework, plant waxes at our core site should reflect changes in the vegetation zones south of the Sahara Desert as well as variation in trade wind strength modulating the amount of waxes transported from the Med source. Strong trade winds would deliver C$_3$ plant waxes and lead to overall more negative δ$^{13}$C$_{wax}$ values with savanna source C$_3$/C$_4$ vegetation changes superimposed. The effect of the Med source would be most pronounced during weak monsoon intervals when there is reduced vegetation cover in the savanna source region, and therefore the Med source makes up a larger proportion of the total wax flux. During periods of greater transport of Med source plant waxes, weak monsoon intervals will have more negative wax δ$^{13}$C values reflecting the increased proportion of Med source C$_3$ waxes.

A pollen reconstruction extending beyond the LGM at ODP 659 (adjacent to our core site) shows much higher fluxes of trade wind indicator pollen between 700 and 300 ka compared to the last ~130 kyrs suggesting stronger trade winds during our study interval[31] of MIS 13–10. This agrees well with the higher dust fluxes, sensitive to trade wind strength, found in this study. These stronger trade winds should increase the flux of C$_3$ plant waxes from the Med source and increase the amplitude of δ$^{13}$C changes. During weak monsoon intervals, the Med source increases the overall C$_3$ contribution to our core site leading to more negative δ$^{13}$C values. During strong monsoon intervals, the effect of increased transport of C$_3$ waxes from the Med source would be muted by the expanded C$_4$ grasslands. The effect of this is an increased amplitude of δ$^{13}$C change without any change in the vegetation response to orbitally driven monsoon changes. Therefore, we expect a stronger relationship between δD$_{precip}$ (a proxy for monsoon intervals) and δ$^{13}$C during MIS 13–10 compared to during the past ~130 kyrs. However, we find the opposite. The relationship between δD and δ$^{13}$C is strongest during the last 130 kyrs and weaker during MIS 13–10.

An alternative scenario, and the one favored here, is that the C$_3$ source to our core site is predominantly the savanna source to the south. The increased C$_3$ component observed during weak monsoon intervals is not the result of increased transport of waxes from the Med source, but rather the result of the reduced area of C$_4$ grasses. Further evidence for this interpretation comes from pollen reconstructions of the latitude of the Sahara/Sahel and savanna/rainforest transitions which show more northerly excursions during MIS 13–10 by several degrees of latitude compared to the last glacial cycle[83]. During MIS 13–10 when

ecosystem boundaries were farther northward, the wax-shed of our core site would have been composed of a smaller area of desert ecosystems and a greater area of grasslands, mixed woody/grassy savanna, and Sudanian deciduous forests. At this time, increased rainfall would have led to expanded grasslands increasing C$_4$ plants, but also increased woody cover in savannas and to a lesser degree expanded deciduous forests which would cause a simultaneous rise in C$_3$ plants in the wax-shed resulting in a more muted increase in the C$_4$ character of the waxes than during intervals when ecosystem boundaries were farther southward (e.g., the last ~130 ka). An additional consequence of these more northerly ecosystem locations is that a greater percentage of the wax-shed is constituted by mixed woody/grassy savannas, making the δ$^{13}$C record of this interval particularly sensitive to vegetation changes in these ecosystems. The differences in variance in the δ$^{13}$C records explained by monsoon rainfall change and atmospheric CO$_2$ concentration can be understood within this framework. During the last ~130 kyrs, the Sahara Desert extended farther southward. Thus, the wax-shed was more strongly influenced by changes in the northward extend of grassland shifting into the desert during times of increased rainfall and secondarily by changes in the woody cover in savannas resulting from glacial/interglacial changes in atmospheric CO$_2$ and rainfall. During MIS 13–10, the Sahara Desert did not extend as far south and mean position of all biomes was farther north. Thus, the wax shed was more strongly influenced by changes in the woody/grassy balance within savanna ecosystems than the northward extent of grasslands. While we cannot rule out some contribution of Med source plant waxes to our study site as indicated by the presence of Mediterranean pollen at these latitudes in the modern and paleo-record, it is unlikely that these waxes make up the bulk of the waxes transported to the core site during this interval. We, therefore, interpret our plant-wax isotopic reconstructions as representing paleoecological and paleohydrologic changes south of the Sahara Desert.

The hydrogen isotope composition (δD) of plant waxes is more directly linked with changes in local hydrological conditions because the source of hydrogen in wax compounds is local precipitation waters[84] which have been shown to be dominantly controlled in this region via the amount effect: Rayleigh fractionation leads to more depleted hydrogen isotopes in rainwater as rainfall increases[85–88]. Biosynthesis of plant waxes imparts an additional apparent fractionation ($\epsilon_a$) on the hydrogen isotopic composition of plant waxes relative to the source water that is dependent on the plant photosynthetic pathway[84]. To calculate the hydrogen isotopic composition of precipitation (δD$_{precip}$) the apparent fractionation factor ($\epsilon_a$) for each sample must be determined based on the fraction of C$_3$ and C$_4$ photosynthesizers present in the landscape. The fraction of C$_4$ contribution to our C$_{31}$ n-alkane record was calculated using a linear mixing model between δ$^{13}$C values of the C$_{31}$ n-alkane for C$_4$ (δ$^{13}$C$_{C_4}$ = −19.9‰) and C$_3$ (δ$^{13}$C$_{C_3}$ = −33.6‰) plant end members for the African continent calculated from a compilation of measurements of modern plants[77,89–92] following:

$$f_{C_4} = \left(\delta^{13}C_{meas} - \delta^{13}C_{C_3}\right)/\left(\delta^{13}C_{C_4} - \delta^{13}C_{C_3}\right) \qquad (2)$$

The apparent fractionation factor ($\epsilon_{a,landscape}$) hydrogen isotopic composition of precipitation is then calculated as an average of the $\epsilon_a$ of C$_3$ and C$_4$ vegetation weighted by their relative contributions to a given sample:

$$\epsilon_{a,landscape} = f_{C_3}\epsilon_{a,C_3} + f_{C_4}\epsilon_{a,C_4} \qquad (3)$$

Where $f_{C_3}$ is calculated as $1 - f_{C_4}$ and $\epsilon_{a,C3}$ and $\epsilon_{a,C4}$ are taken from $\epsilon_a$ values for C$_3$ angiosperm trees and C$_4$ grasses in ref. [84]. δD$_{precip}$ can then be calculated as:

$$\delta D_{precip} = (\delta D_{wax} + 1)/(1 + \epsilon_{a,landscape}) - 1 \qquad (4)$$

Here we interpret the calculated δD$_{precip}$ as qualitatively representative of changes in the amount of monsoonal precipitation (i.e., wet season rainfall).

Recent work has shown that orbital scale variability in winter season rainfall in the Mediterranean region likely contributed to the northward expansiveness of grassland and savanna ecosystems during Green Sahara episodes[93], however the δD$_{precip}$ in North Africa during humid intervals is ~10‰ more positive than at our core site[18], therefore any contribution of waxes to our study site from this source would act to slightly dampen the observed δD$_{precip}$ monsoon signature. During arid intervals when pollen suggests an increased northerly source, δD$_{precip}$ values are similar in both locations, suggesting a minimal effect on the δD$_{precip}$ record. Thus, any Med source contribution would act to only slightly dampen the amplitude of the δD$_{precip}$ record and thus inferred monsoon rainfall variability. We, therefore, as for our carbon isotope record, interpret the δD$_{precip}$ variability to primarily reflect changes in wet season rainfall in the grasslands, savannas, and deciduous forests south of the Sahara, and to a much lesser extent the steppe and Mediterranean forests north of the desert on orbital timescales.

**Assessing evidence of hydrologic change during the Last Glacial Maximum.** Proxy records of the hydrological conditions during the LGM (~23–19 ka) in western equatorial and northern Africa generally consist of reconstructions either directly sensitive to changes in precipitation relative to evaporation (P–E) or indirectly sensitive to continental aridity via inferred relationships between terrestrial vegetation and hydroclimate. Along the northwest African margin, marine core proxy records of continental rainfall derived from plant-wax δD, show similar (and in some locations more negative) values of precipitation hydrogen isotopes from ~15–31°N between the LGM and the late-Holocene[18,42,94] indicating similar

if not more intense monsoon rainfall during the LGM. Lake level reconstructions from across the Sahara Desert—the most direct evidence for changes in P−E— reveal a similar pattern of comparable or wetter LGM conditions. A larger fraction of lakes displayed high or medium lake levels during the LGM compared to the late-Holocene[62]. Terrestrial dust fluxes to marine sediments off of the Sahara, sensitive to aridity and wind strength, also show similar values between the LGM and late Holocene[20,21,95], which have been previously attributed[96] to increased soil moisture in Northwest Africa resulting from reduced evaporative demand during the LGM[38]. Farther south, marine core records from along the tropical West African margin as well as lacustrine records sensitive to western topical African rainfall show a more complicated pattern of LGM hydroclimate. Sea surface salinity (SSS) reconstructions based on planktic foraminifera Ba/Ca ratios suggest ~1.5 psu higher SSS in the vicinity of the Niger and Sanaga Rivers (~2.5°N) during the LGM implying reduced runoff and thus rainfall[97]. However, decreased salinity during the LGM in the eastern Gulf of Guinea compared to the late Holocene may also be in part due to a more saline LGM ocean resulting from greatly expanded Northern Hemisphere ice sheets (equivalent to ~+1 psu by sea water mass balance). Offshore of the Congo River basin (~6°S) planktic foram-derived $\delta^{18}O_{SW}$ (an SSS proxy) and plant-wax $\delta D$ records show negligible differences between the LGM and late Holocene[98,99]. Overall, the direct hydroclimate proxies indicate similar or wetter LGM conditions in northern and tropical West Africa compared to the late Holocene, while only one record of SSS from the eastern Gulf of Guinea suggests rainfall may have been marginally lower.

In contrast, indirect inferences of rainfall from vegetation-based proxies are often interpreted to indicate drier conditions during the LGM. Marine[100–102] and lacustrine[103–106] sediment core pollen and bulk sediment $\delta^{13}C$ records from ~7°N to ~10°S show an equatorward contraction of forest ecosystems and expanded savanna cover during the LGM compared to the late Holocene. Other extensive pollen compilations of primarily terrestrial sites in the Congo basin show the same pattern[107,108]. The classical interpretation of these vegetation reconstructions is that forest replacement by savannas during the LGM indicates decreased rainfall, based upon the modern relationship between savanna distributions and rainfall patterns. However, this interpretation neglects the effect of $CO_2$ concentrations on biome distributions. Previous authors do note that the effect of $CO_2$ concentrations is a large source of uncertainty[109]. Vegetation models that include the physiological effects of low $CO_2$ levels during the LGM on $C_3/C_4$ competition have convincingly demonstrated that when $CO_2$ effects are ignored, biome-based reconstructions of LGM aridity are significantly biased toward dry conditions[110–113]. Therefore, much of the vegetation-based evidence of a dry LGM in tropical West Africa is explained by low $CO_2$ levels rather than decreased rainfall. Accounting for $CO_2$ effects brings vegetation-based estimates in line with proxy and model predictions of little change in P−E in this region during the LGM[38,39]. In this study, we, therefore, conclude that rainfall in northern and tropical West Africa during the LGM was broadly similar to the late Holocene.

**Dust flux calculations and He isotope analyses.** Dust fluxes were determined by multiplying the terrestrial fraction of the sediments with the mass flux (mass accumulation rate). We constrain the mass flux record using extraterrestrial helium-3 ($^3He_{ET}$) as a constant flux proxy in refs. [114–116]. Traditionally, sedimentary flux records are derived from stratigraphic accumulation rates that rely on the mass of sediments accumulated between age model tie-points. However, these age model-derived flux records must assume constant mass fluxes between age model tie points and may be inflated or deflated by the effects of sediment winnowing and focusing, respectively[117]. In contrast, utilization of a constant flux proxy allows for point-by-point determination of vertical sediment fluxes that are independent of lateral sediment transport.

Helium isotopes were measured in carbonate-free sediment samples. To remove calcium carbonate, sediment samples were first leached in 25–30 ml of 10% acetic acid under light agitation overnight. Acidic solutions were then rinsed with mQ water and centrifuged for 5–10 min at 1370 g three times. Samples were then left to evaporate overnight before freeze-drying. The helium was extracted from freeze-dried samples via heating in a furnace at ~1300 °C and measured on an MAP-215 mass spectrometer[115]. Helium isotopes found in marine sediments represent a combination of terrestrial and extraterrestrial input. Measured $^3He$ concentrations were converted into mass accumulation rate estimates using a binary end-member mixing model using an Interplanetary dust particle (IDP) extraterrestrial end-member $^3He/^4He$ ratio of $2.4 \times 10^{-4}$ ref. [118], an IDP $^3He$ flux of $8 \times 10^{-13}$ cc/cm²/kyr ref. [116], and a terrestrial end-member $^3He/^4He$ ratio of $1 \times 10^{-8}$ (see next section for calculation). Extraterrestrial $^3He$ concentrations are determined using a two-component mixing model between the $^3He/^4He$ concentration ratio of the terrestrial and extraterrestrial source material following Eq. (5).

$$[^3He_{ET}](\text{pcc g}^{-1}) = [^3He_{meas}](\text{pcc g}^{-1}) * \left\{ \frac{1 - \frac{^3He/^4He_{Terr}}{^3He/^4He_{meas}}}{1 - \frac{^3He/^4He_{Terr}}{^3He/^4He_{IDP}}} \right\} \quad (5)$$

The $^3He/^4He$ ratio of the extraterrestrial interplanetary dust particles (IDPs) end member is spatially consistent and taken to be $2.4 \times 10^{-4}$ ref. [118]. However, the terrestrial end member $^3He/^4He$ ratio has been shown to vary considerably with dust source region[119]. Here we model the terrestrial end member $^3He/^4He$ ratio using a Monte Carlo approach (see next section for details).

Mass accumulation rate (MAR) is calculated following Eq. (6)

$$\text{MAR}(\text{g cm}^{-2}\text{kyr}^{-1}) = {}^3He_{IDP}\text{Flux (pcc cm}^{-2}\text{ kyr}^{-1})/[^3He_{ET}](\text{pcc g}^{-1}) \quad (6)$$

Where IDP flux is taken from ref. [116] and $^3He_{ET}$ is determined by Eq. (5). The dust flux is then calculated by multiplying the mass accumulation rate by the terrestrial weight fraction (1 − fraction carbonate − fraction opal) (Eq. 7).

$$\text{Dust flux}(\text{g cm}^{-2}\text{ kyr}^{-1}) = \text{MAR}(\text{g cm}^{-2}\text{ kyr}^{-1}) * (1 - F_{CaCO3} - F_{Opal}) \quad (7)$$

Uncertainty estimates for the $^3He_{ET}$-normalized dust flux were computed by propagating the $^4He$ and $^3He$ concentration analytical (air standard He concentration uncertainty $1\sigma = 1\%$), replicate concentration fractional difference ($^3He$ $1\sigma = 22.9\%$, $^4He$ $1\sigma = 16.6\%$; Fig. S2), a nominal 5% uncertainty in the fraction terrestrial estimate for all sediment samples, and a $1\sigma = 40\%$ uncertainty in the IDP $^3He$ flux from ref. [116].

**Terrestrial end member $^3He/^4He$ modeling and dust flux sensitivity.** Five samples (Supplementary Data 4) from this core were analyzed for both He and Th concentrations following the methods described in the main text and previously[20]. $^3He_{ET}$ and $^{230}Th_{XS}$-derived mass accumulation rates (MARs) should have a regression slope of $m = 1$ if the terrestrial end member $^3He/^4He$ ratio is correct. Here we test how the choice of different terrestrial endmembers affects the slope of the total least squares fit between $^{230}Th_{XS}$- and $^3He_{ET}$-derived mass accumulation rates. A total of 500 evenly spaced possible terrestrial end member $^3He/^4He$ ratio values were tested over the range of $1 \times 10^{-9}$–$1 \times 10^{-7}$ mol/mol where for each potential end member ratio $n = 10,000$ Monte Carlo artificial $^3He_{ET}$- v $^{230}Th_{XS}$-MAR regressions were calculated by randomly sampling the measured Th and He concentrations and the IDP flux from a normal distribution defined by the published mean and $1\sigma$ uncertainty[116] and randomly sampling a uniform distribution of IDP end member $^3He/^4He$ ratio values between $1–4 \times 10^{-4}$. For each of the 500 possible terrestrial end members, a probability density function (PDF) of MAR regression slopes was created. To assess the likelihood that any given end member ratio resulted in the best fit, the value of these PDFs at a slope of one was extracted and plotted as a function of the terrestrial $^3He/^4He$ end member value (Fig. S10). For end member values less than ~$1.5 \times 10^{-8}$ the probability the Th/He MAR regression slope is equal to one is consistently high (>0.85), however the probability rapidly declines for larger end member $^3He/^4He$ ratios. Here we choose a terrestrial end member value of $1 \times 10^{-8}$ to calculate $^3He_{ET}$ mass accumulation rates. This estimate agrees with $^3He/^4He$ ratios of terrestrial soil samples from across North Africa[120] which all measured below $1.4 \times 10^{-8}$. Future work including additional paired measurements of Th- and He-derived mass accumulation rates or direct measurements of the $^3He/^4He$ ratio of Saharan surface samples would help refine the uncertainty on this end member value.

While the timing and thus periodicity of the determined dust fluxes are not impacted by terrestrial end member choice, because the $^3He_{ET}$-derived MAR is a nonlinear function of the terrestrial $^3He/^4He$ end member ratio, the amplitude of total dust flux variability increases for larger terrestrial $^3He/^4He$ end member ratios. To assess the sensitivity of our derived dust fluxes to the choice of terrestrial end member we calculated the $^3He_{ET}$ constant flux normalized dust fluxes over a wide range of end member $^3He/^4He$ ratios (Fig. S11). Constrained by the Monte Carlo simulation described above and by previous estimates of North African soil $^3He/^4He$ ratios all consistent with $^3He/^4He$ ratios <$1.4 \times 10^{-8}$ ref. [120], the end member value is unlikely to be higher than the conservative upper bound on the $^3He/^4He$ terrestrial end member ratio of $2.0 \times 10^{-8}$, which when used to calculate dust flux results in all fluxes, save the two highest (~390 and ~455 Ka), falling within uncertainty of fluxes calculated using the ratio of $1.0 \times 10^{-8}$ determined by the Monte Carlo procedure outlined above (Fig. S10). The lower bound, however, is not well constrained. While the uncertainty in the amplitude of the dust flux variability does not impact our conclusions about the timing or periodicity of dust flux changes in our record, it does add an additional uncertainty when comparing with other dust flux records that use a different normalization technique, e.g., $^{230}Th_{XS}$ normalization. Future work to better constrain the terrestrial end member $^3He/^4He$ ratio in North Africa either by improving the intercalibration between the $^{230}Th_{XS}$ and $^3He_{ET}$ techniques or by refinement of the terrestrial end member $^3He/^4He$ ratio via direct surface sediment samples is needed to aid future comparisons between dust fluxes derived by these two normalization techniques. Here, we take a conservative approach and do not interpret differences in the absolute value of dust flux determined by different normalization techniques.

**Statistical techniques.** Linear regressions were determined by first linearly interpolating higher resolution environmental and orbital records to the ages of the records generated in this study, then correlation coefficients were calculated using the "fitlm" command in MATLAB v2020a, corresponding p-values were calculated following ref. [121] which adjusts for reduced degrees of freedom resulting from serial autocorrelation of related time series. For wavelet and cross-wavelet analysis, we follow the procedures proposed by refs. [122–125] respectively employing a Morlet mother wavelet (wavelet parameter = 5) with all time series linearly interpolated to 3 kyr intervals. Insolation time series were generated using MATLAB package "daily_insolation" ref. [126] following the insolation solutions of refs. [126,127].

## Data availability

The data generated in this study are provided in Supplementary Data 1–4.

## Code availability

All code used in this analysis has been previously published and has been appropriately cited throughout the text.

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

## Acknowledgements

IMAGES core MD03-2705 was recovered by the R/V Marion Dufresne (Institut Paul Emile Victor) in 2003. We thank Jessica Tierney, Tom Johnson and one anonymous referees for their comments. These authors would also like to acknowledge the National Science Foundation (award #1502925) for funding and Nicole deRoberts, Helen Habicht, and Linda Baker for laboratory assistance. C.S. was partly supported by the French national program LEFE/INSU. G.W. acknowledges support from the Vetlesen Foundation.

## Author contributions

N.A.O. and C.S. performed the analyses; D.M., G.W., P.J.P., and C.S. conceived the study, N.A.O. wrote the initial draft of the manuscript, and N.A.O., C.S., D.M., G.W., A.J.M.B., L.I.B., B.M., and P.J.P. wrote the final version of the manuscript.

## Competing interests

The authors declare no competing interests.
