## [Peer Review File · Nature Communications]

Pleistocene drivers of Northwest African hydroclimate and vegetationReviewers' Comments:

Reviewer #1:

Remarks to the Author:

O'Mara et al present a new dataset of leaf wax δD , $\delta^{13}C$, and dust flux from a marine sediment core spanning MIS 13-MIS 11. The δD and dust data show a close relationship to precession, while the $\delta^{13}C$ data follow atmospheric CO_2 concentrations. The authors conclude that low-latitude insolation gradients drive changes in the African monsoon, while CO_2 concentrations dominate the trade off b/t C3 and C4 ecosystems in N Africa.

This is a really excellent new dataset and a nice complement to the existing leaf wax data from ODP 659 (Kuechler et al 2013). The precessional flavor of the leaf wax δD data is compelling, as is the very close match b/t $\delta^{13}C$ and CO_2 . I've just got a few suggestions of how to strengthen some of the arguments in the paper.

1) C4 grasses (and grasses in general) do have a substantially different hydrogen isotope apparent fractionation, so usually a correction is made for this, to adjust δD_{wax} for this overprint. I see that the authors did do this - it is described in the Supplement - and the corrected versions appear in ED Figure 10. But they do not show the calculated δDP in Figure 1 nor do they analyze the spectral characteristics. To my eye, it looks like the corrected data are not that different, but to be thorough I think it is best to analyze the computed δDP values (the best guess as to what the isotopes of rainfall were) for the spectral characteristics and compare to the insolation curves, etc. In other words, δDP should be in the main text figures, not the raw δD_{wax} . This way, the impact of vegetation change is accounted for before the δD variability is interpreted in the main text.

2) I would like to see a little more mechanistic discussion surrounding the links b/t insolation and the monsoon proxies. There is a lot of focus on "insolation gradients" in the paper, particularly the hemispheric gradient when the precession cycle is amplified. However, I'm not sure I understand why a "gradient" is a more likely explanation than just local insolation changes. Matching up insolation curves and looking at the spectra doesn't answer this question, since there are physically unique solutions that could equally explain the match, so interpretations should be grounded in dynamics. Mechanistically, the monsoon responds to the availability of moist static energy, which fuels the monsoon low, and this is directly related to incoming solar radiation in the early summer (June). Does the southern hemisphere energy budget actually matter? Bosmans et al. 2015 (Climate Dynamics) suggest that the south Atlantic high intensifies and therefore helps fuel cross-equator winds, but is this a response to local (SH) insolation directly or actually the NH changes? I'm not super familiar with the paper, so I'm not sure. I guess I would explain further in the manuscript here why a "gradient" in insolation is needed vs. just a response to more energy in the NH summer.

Likewise, with obliquity, beyond a direct effect discussed in the Bosmans papers, there is a possibility it affects the monsoon systems indirectly through ice-sheet-induced ventilation. (c.f. Chou and Neelin, 2003; Bhattacharya et al., 2018, and DiNezio et al., 2018). In the present day, the African system is less sensitive to this mechanism (see Chou and Neelin) than the North American and Asian systems, but I'm not sure whether this situation changes over G/IG timescales. This would be a way to transmit the obliquity signal to the tropics, i.e. through growth of Eurasian ice, limiting monsoonal extent, and could be an alternative explanation to the obliquity-related interhemispheric gradient. If you think this a less likely explanation, that's fair, but it should be considered and discussed.

I would also throw out there that one reason that the Saharan region in particular might not "feel" the ice sheets is that it is too dry. I.e. the Sahara is really dry today and so cooler climate, ice sheet effects can't do much. Basically, a threshold issue. Indeed, leaf wax records elsewhere from Africa (Gulf of Aden, Tanganyika) do show 100 ka power. I wonder if a long record from farther south (sahel/tropical interface) would show something different. You might want to discuss this - that the Sahara is particular might be very precessional b/c the ice ages just can't dry the place out further.

3) I would be a bit more cautious about extrapolating a calculation of the increase in woody cover from the glacial/interglacial past to a high CO₂ future. The G/IG cycles are covering a really low range in CO₂. 190 ppm very strongly selects for C₄. But this isn't the case moving forward under a 400+ ppm world. Arguably, C₃ is already favored at current CO₂ levels so a further increase has less of an effect than what is seen over G/IG cycles. Hence using a G/IG relationship and carrying it forward into the future will be an overestimation. Moreover, the CO₂ fertilization effect might be a temporary phenomenon, again b/c once one gets to high enough CO₂ it's not as effective. See Bathiany et al., 2014 for a nice discussion of this (<https://doi.org/10.1175/JCLI-D-13-00528.1>). So I would not go so far as to predict a massive greening. And certainly, this discussion should reference some more papers studying present and future Sahel greening.

Some additional line-by-line comments are listed below. Nice work!

-Jess Tierney

Line 75: this mention of the LGM is a little out of place as presumably the low CO₂ of the LGM would favor C₄ expansion rather than C₃?

Line 126: Here are elsewhere: correlations should be given as r values not r^2 , which is the coefficient of determination and refers to regression. All p values should be adjusted for reduced degrees of freedom due to serial correlation, i.e. through a non-parametric method like that of Ebisuzaki et al., 1997 ([https://doi.org/10.1175/1520-0442\(1997\)010<2147:AMTETS>2.0.CO;2](https://doi.org/10.1175/1520-0442(1997)010<2147:AMTETS>2.0.CO;2)) or another approach. I can give you Matlab code to do this if you want - just send me an email.

Line 166-174: Lower rainfall does not mean more C₄ grass. C₄ grass needs some intermediate amount of rain, usually about 500 mm/yr, to grow. Once you drop below that, the grass disappears from the landscape. This is the case in the arid Horn of Africa for example (Liddy et al., 2016 makes this point: <https://www.sciencedirect.com/science/article/pii/S0012821X1630214X>) and here in southern Arizona. In the Sonoran desert we get about 300 mm of rain each year and there is very little grass. Just to the south, go up-elevation a bit and annual rainfall jumps up - C₄ grass is suddenly everywhere! In this context, increased C₄ during wet periods makes total sense, b/c a strong monsoon is allowing the grassland to expand into the desert. I think that is what you are hinting at towards the end of the paragraph and in the next one as well, but I would just make this more clear and add some citations to the grassland precip thresholds.

Reviewer #2:

Remarks to the Author:

Review of 'Past and future drivers of Northwest African hydroclimate and vegetation' by N. A. O'Mara et al.

The manuscript consists of 3 relatively short parts which are not directly connected to each other and provide relatively little discussion of the suggested new findings. The first part is on the continental hydroclimatic evolution of a 180 kyr interval (from 530 to 350 ka ago) inferred from a marine core (MD03-2705) which is compared to existing hydroclimatic reconstructions from a nearby location (ODP659) from the last glacial cycle and the Pliocene and comes with findings on the drivers of hydroclimatic evolution of northwest Africa. The second part is about dust flux estimates for the same time interval which are proposed as aridity indicator. The last part is about a proposed CO₂ effect on vegetation composition in African savannahs. Each of these 3 parts builds heavily on prior published data and re-iterates ideas that have previously been discussed in the literature while presenting only limited new data and ideas. I do not see the novelty of the findings and interpretations and therefore

recommend rejection of the manuscript. If there are indeed novel aspects these should preferentially be published in journals allowing more discussion. Specific comments for each part in the following:

Hydroclimate drivers of NW Africa:

For the assessment of rainfall variations at the core site compound-specific stable hydrogen isotope compositions (δD) of leaf waxes are used as measure of rainfall amount (should be better: rainfall intensity during the growing season). The core site is located offshore the modern boundary between the Saharan desert and the Sahelian grass savannah. It is assumed that wax transport is mainly from east to west. Additionally, the site may be influenced by material transported from the north via the trade winds. It is common to find Mediterranean pollen down to this latitude. The source area of the waxes is not properly discussed. If any material is transported from the northern Mediterranean areas, what would the impact on the δD (and $\delta^{13}C$) records be? Implicitly it is assumed that the monsoon is the only moisture source in the area. It has recently been shown that the northern part of NW Africa is influenced by increased winter rainfall contemporaneous to an intensified monsoon during insolation maxima with a possibly overlapping zone of 2 rainfall regimes in the Sahara (Cheddadi et al., 2021, PNAS). With 2 interfering moisture sources carrying different isotope compositions any conversion of δD into rainfall amounts as done in this manuscript (and other studies before) cannot be done. Moisture sourced from the mid-latitude Atlantic likely explains the obliquity component in the hydroclimatic forcing. The idea of insolation gradients as drivers of hydroclimate changes in NW Africa is presented and extensively discussed in Kuechler et al. (2018, *Climate of the Past*). While this manuscript now comes to slightly different conclusions than Kuechler et al. (2018) (interhemispheric tropical versus latitudinal insolation gradient), it reads as if the idea presented here is new. If the slightly different findings to Kuechler et al. (2018) warrants a dedicated publication they should be discussed in more detail in a more specialized journal.

Dust fluxes:

Aeolian dust needs deflatable source areas AND strong winds to result in elevated dust fluxes to offshore areas. If either of the components is not given, no elevated dust fluxes will be created. As extensively discussed in Trauth et al. (2009, QSR) dust therefore mainly is an 'aridification proxy' indicating deflation of previously vegetated areas under strong wind conditions. It is not a measure of aridity itself as under prolonged drought conditions dust source areas will be depleted. The authors present new estimates of dust fluxes based on proxy-normalization which show a strong anti-correlation with inferred rainfall intensity contradicting the earlier view of dust generation and transport. No discussion is made about the significance of the new dust proxy. Findings in this manuscript are repeated from Skonieczny et al. (2019, *Science Advances*) presenting data from the same sediment core. I cannot see how this part of the manuscript is adding new insights nor is connected to the prior or following part of discussion.

C3/C4 occurrence related to pCO_2 :

Since the Pliocene the broad distinction of African photosynthetic vegetation types are C3 trees and C4 (warm-season) grasses. C3 (cold-season) grasses only occur in specific areas such as the Mediterranean coast and the western Cape receiving substantial amounts of winter rainfall. The occurrence of trees and shrubs (C3) is limited by the length of the dry season. A long dry season will not allow trees and shrubs to survive. That is why the length of the dry season is the primary control of the occurrence of woody vegetation (C3) in modern African savannahs (see supplement to Collins et al., 2011, *Nature Geoscience*). With no large pCO_2 change from the mid- to Late Holocene (excluding modern data) a large C3 increase is detected in savannahs of both Hemispheres (Collins et al., 2011) testifying that this principle also holds true in the past. For the LGM, a C4 increase is detected in savannahs on both hemispheres (Collins et al., 2011) correlating to vast evidence for overall drier conditions during the LGM (e.g., Gasse et al., 2000, QSR). The authors now propose (based on the quantum yield model from Ehleringer et al., 1997, *Oecologia*) that "the physiological control of atmospheric pCO_2 on photosynthesis dominantly controls the woody/grassy balance in existing savannahs". This is a re-iteration of arguments proposed by Bragg et al. (2013, *Biogeosciences*) who used the same re-interpretation of data originally presented by Collins et al. (2011). Obviously, the

pCO₂ effect on C₃/C₄ would be an alternative explanation of the C₄ increase during the LGM but only if total rainfall (or wet season length) remained the same. Ascribing the entire C₄ increase during the LGM to a pCO₂ effect as done by the authors (Fig. 4) is highly unrealistic in the light of independent data signaling increased aridity. Why is there a C₃ increase at the LGM in the equatorial rainforest? This speaks against a control by pCO₂. The authors' statement cited above is thus not true. Obviously, this consideration does not rule out that there may be an effect of 'woody thickening' in tree savannahs due to CO₂ fertilization under otherwise relatively stable hydroclimatic conditions. This effect, however, is difficult to disentangle in modern times from land-use changes or changes in agricultural/farming practice. The data from the past (with a dominant control of wet season length on C₃/C₄) do not help to forecast pCO₂ effects.

Other comments:

I wonder how the authors disentangle pCO₂ changes from changes in global ice volume (which are tightly coupled to each other, e.g. Petit et al., 1999, Nature). Increases in global ice volume act to symmetrically increase the latitudinal temperature gradient, compressing the Hadley circulation which restricts the latitudinal migration of the rainbelt and thus lead to decreased wet season length in both hemispheres.

The authors' argument that "During the LGM...rainfall as estimated by plant-wax dD was broadly similar to modern" ignores decades of work by other disciplines (e.g., Gasse, 2000) showing a generally dry LGM in Africa. Besides, it is a mis-understanding as dD of leaf waxes reflects rainfall intensity during the growing season and tells nothing about mean annual rainfall amount. The dD data corresponding to the time-slice 13C data (Collins et al., 2013, QSR) show a southward shift in rainfall intensity (but not rainfall amounts) during the LGM which is de-coupled from the change in wet season length.

The 'surprising' finding of C₄ grass expansion into the Sahara Desert during wet phases is already described and explained by Kuechler et al. (2013).

The quantitative conversion of dD_{wax} to dD_{precip} as proposed by Niedermeyer et al (2016, GCA) which is used in this manuscript is valid only for the Holocene and large uncertainties arise for other times due to the stability of the amount effect under different boundary conditions, other interfering moisture sources, etc. This is clearly noted in Niedermeyer et al. (2016).

Response to reviewers for Nature Communications manuscript NCOMMS-21-15793-T: "Past and future drivers of Northwest African hydroclimate and vegetation"

Reviewer #1 (Remarks to the Author):

O'Mara et al present a new dataset of leaf wax δD , $\delta^{13}C$, and dust flux from a marine sediment core spanning MIS 13-MIS 11. The δD and dust data show a close relationship to precession, while the $\delta^{13}C$ data follow atmospheric CO_2 concentrations. The authors conclude that low-latitude insolation gradients drive changes in the African monsoon, while CO_2 concentrations dominate the trade off b/t C3 and C4 ecosystems in N Africa.

This is a really excellent new dataset and a nice complement to the existing leaf wax data from ODP 659 (Kuechler et al 2013). The precessional flavor of the leaf wax δD data is compelling, as is the very close match b/t $\delta^{13}C$ and CO_2 . I've just got a few suggestions of how to strengthen some of the arguments in the paper.

1) C4 grasses (and grasses in general) do have a substantially different hydrogen isotope apparent fractionation, so usually a correction is made for this, to adjust δD_{wax} for this overprint. I see that the authors did do this - it is described in the Supplement - and the corrected versions appear in ED Figure 10. But they do not show the calculated δD_{precip} in Figure 1 nor do they analyze the spectral characteristics. To my eye, it looks like the corrected data are not that different, but to be thorough I think it is best to analyze the computed δD_{precip} values (the best guess as to what the isotopes of rainfall were) for the spectral characteristics and compare to the insolation curves, etc. In other words, δD_{precip} should be in the main text figures, not the raw δD_{wax} . This way, the impact of vegetation change is accounted for before the δD variability is interpreted in the main text.

Thank you for this suggestion, the figures and text throughout the manuscript have been updated to reflect δD_{precip} (i.e. ice volume and vegetation corrected values) rather than just the ice volume corrected δD_{wax} values that were in the previous draft.

2) I would like to see a little more mechanistic discussion surrounding the links b/t insolation and the monsoon proxies. There is a lot of focus on "insolation gradients" in the paper, particularly the hemispheric gradient when the precession cycle is amplified. However, I'm not sure I understand why a "gradient" is a more likely explanation than just local insolation changes. Matching up insolation curves and looking at the spectra doesn't answer this question, since there are physically unique solutions that could equally explain the match, so interpretations should be grounded in dynamics. Mechanistically, the monsoon responds to the availability of moist static energy, which fuels the monsoon low, and this is directly related to incoming solar radiation in the early summer (June). Does the southern hemisphere energy budget actually matter? Bosmans et al. 2015 (Climate Dynamics) suggest that the south Atlantic high intensifies and therefore helps fuel cross-equator winds, but

is this a response to local (SH) insolation directly or actually the NH changes? I'm not super familiar with the paper, so I'm not sure. I guess I would explain further in the manuscript here why a "gradient" in insolation is needed vs. just a response to more energy in the NH summer.

We appreciate this comment highlighting a lack of clarity on this topic in the manuscript. We have refined the language in the " δD_{precip} reveals dynamics of Northwest African monsoon insolation forcing" section of our manuscript to more explicitly describe why insolation gradients are required to explain the reconstructed monsoon variability. Specifically see a newly added paragraph at the end of that section for our clarified interpretation of the influences of both changes in direct solar heating and gradient-driven moisture advection on the monsoon rainfall response.

Likewise, with obliquity, beyond a direct effect discussed in the Bosmans papers, there is a possibility it affects the monsoon systems indirectly through ice-sheet-induced ventilation. (c.f. Chou and Neelin, 2003; Bhattacharya et al., 2018, and DiNezio et al., 2018). In the present day, the African system is less sensitive to this mechanism (see Chou and Neelin) than the North American and Asian systems, but I'm not sure whether this situation changes over G/IG timescales. This would be a way to transmit the obliquity signal to the tropics, i.e. through growth of Eurasian ice, limiting monsoonal extent, and could be an alternative explanation to the obliquity-related interhemispheric gradient. If you think this a less likely explanation, that's fair, but it should be considered and discussed.

I would also throw out there that one reason that the Saharan region in particular might not "feel" the ice sheets is that it is too dry. I.e. the Sahara is really dry today and so cooler climate, ice sheet effects can't do much. Basically, a threshold issue. Indeed, leaf wax records elsewhere from Africa (Gulf of Aden, Tanganyika) do show 100 ka power. I wonder if a long record from farther south (sahel/tropical interface) would show something different. You might want to discuss this - that the Sahara is particular might be very precessional b/c the ice ages just can't dry the place out further.

Thank you for providing these references and a possible mechanism to explain a lack of clear ice volume influence on the monsoon-sensitive proxies we reconstructed here. Upon closer inspection of our data we find that the obliquity component of the δD_{precip} significantly leads that of the global ice volume, precluding ice sheet-induced ventilation as the direct cause of the obliquity variability in the Northwest African monsoon. To show this, we have added a new Extended Data figure (fig. ED7) and included a further discussion of this ventilation mechanism to the main text. As an aside, I would also like to point out that our conclusions do not weigh in on the applicability of this ventilation mechanism in rainfall patterns for other parts of the continent, e.g. from the Horn of Africa where 100kyr power is quite apparent, or elsewhere—we are careful in the text to make this distinction clear. With regard to your question about the Sahel/tropical interface, we have colleagues at Lamont working on similar reconstructions from farther south that will be able to provide insight into this in the near future.

3) I would be a bit more cautious about extrapolating a calculation of the increase in woody cover from the glacial/interglacial past to a high CO₂ future. The G/IG cycles are covering a

really low range in CO₂. 190 ppm very strongly selects for C₄. But this isn't the case moving forward under a 400+ ppm world. Arguably, C₃ is already favored at current CO₂ levels so a further increase has less of an effect than what is seen over G/IG cycles. Hence using a G/IG relationship and carrying it forward into the future will be an overestimation. Moreover, the CO₂ fertilization effect might be a temporary phenomenon, again b/c once one gets to high enough CO₂ it's not as effective. See Bathiany et al., 2014 for a nice discussion of this (<https://doi.org/10.1175/JCLI-D-13-00528.1>). So I would not go so far as to predict a massive greening. And certainly, this discussion should reference some more papers studying present and future Sahel greening.

This point is well taken. We have restructured the future implications section to more explicitly state that we are drawing conclusions about what the paleorecord is telling us about the relative impact of rainfall and CO₂ change has had on the makeup of the vegetation structure (i.e. C₃/C₄ balance) with the presumption that the response is linear, and represents the “natural” response to these climate perturbations. We go on to explain that the degree to which these forcing result in the predicted natural response will likely be decided in large part as a result of decisions made by humans. We have also removed the explicit calculation of changes in percent woody cover.

Some additional line-by-line comments are listed below. Nice work!

-Jess Tierney

Thank you for the very constructive comments!

Line 75: this mention of the LGM is a little out of place as presumably the low CO₂ of the LGM would favor C₄ expansion rather than C₃?

This sentence was changed to say “*modern vs proxy and model estimates of last glacial maximum (LGM) vegetation...*” to keep the comparison consistent with future greening fertilization experiments and modern trends highlighting shifts towards higher woody cover when atmospheric CO₂ is at higher concentrations.

Line 126: Here are elsewhere: correlations should be given as r values not r², which is the coefficient of determination and refers to regression. All p values should be adjusted for reduced degrees of freedom due to serial correlation, i.e. through a non-parametric method like that of Ebisuzaki et al., 1997 ([https://doi.org/10.1175/1520-0442\(1997\)010<2147:AMTETS<2.0.CO;2](https://doi.org/10.1175/1520-0442(1997)010<2147:AMTETS<2.0.CO;2)) or another approach. I can give you Matlab code to do this if you want - just send me an email.

Thank you for providing the necessary code to update our correlation analysis. Throughout the manuscript r² values were changed to R values, and p-values were recalculated following Ebisuzaki et al., 1997.

Line 166-174: Lower rainfall does not mean more C₄ grass. C₄ grass needs some intermediate amount of rain, usually about 500 mm/yr, to grow. Once you drop below that, the grass disappears from the landscape. This is the case in the arid Horn of Africa for example (Liddy et al., 2016 makes this point:

<https://www.sciencedirect.com/science/article/pii/S0012821X1630214X>) and here in southern Arizona. In the Sonoran desert we get about 300 mm of rain each year and there is very little grass. Just to the south, go up-elevation a bit and annual rainfall jumps up - C₄ grass is suddenly everywhere! In this context, increased C₄ during wet periods makes total sense, b/c a strong monsoon is allowing the grassland to expand into the desert. I think that is what you are hinting at towards the end of the paragraph and in the next one as well, but I would just make this more clear and add some citations to the grassland precip thresholds.

This section has been restructured to make the rainfall limits on C₄ grass vegetation more clear and to describe what the possible important drivers of C₃/C₄ vegetation balance are within these limits and how we address those potential drivers in this analysis.

Reviewer #2 (Remarks to the Author):

Review of 'Past and future drivers of Northwest African hydroclimate and vegetation' by N. A. O'Mara et al.

The manuscript consists of 3 relatively short parts which are not directly connected to each other and provide relatively little discussion of the suggested new findings. The first part is on the continental hydroclimatic evolution of a 180 kyr interval (from 530 to 350 ka ago) inferred from a marine core (MD03-2705) which is compared to existing hydroclimatic reconstructions from a nearby location (ODP659) from the last glacial cycle and the Pliocene and comes with findings on the drivers of hydroclimatic evolution of northwest Africa. The second part is about dust flux estimates for the same time interval which are proposed as aridity indicator. The last part is about a proposed CO₂ effect on vegetation composition in African savannahs. Each of these 3 parts builds heavily on prior published data and re-iterates ideas that have previously been discussed in the literature while presenting only limited new data and ideas. I do not see the novelty of the findings and interpretations and therefore recommend rejection of the manuscript. If there are indeed novel aspects these should preferentially be published in journals allowing more discussion. Specific comments for each part in the following:

Hydroclimate drivers of NW Africa:

For the assessment of rainfall variations at the core site compound-specific stable hydrogen isotope compositions (δD) of leaf waxes are used as measure of rainfall amount (should be better: rainfall intensity during the growing season). The core site is located offshore the modern boundary between the Saharan desert and the Sahelian grass savannah. It is assumed that wax transport is mainly from east to west. Additionally, the site may be influenced by material transported from the north via the trade winds. It is common to find Mediterranean pollen down to this latitude. The source area of the waxes is not properly discussed. If any

material is transported from the northern Mediterranean areas, what would the impact on the dD (and 13C) records be?

Thank you for pointing out the additional uncertainties associated with wax-provenance that we had not adequately addressed in the manuscript. We have expanded our discussion of the possible source areas, or the “wax-shed” of our study site in the methods section of the manuscript with consideration of both the impact on the $\delta^{13}\text{C}_{\text{wax}}$ and $\delta\text{D}_{\text{precip}}$ records presented in this study—please see the text for details.

Implicitly it is assumed that the monsoon is the only moisture source in the area. It has recently been shown that the northern part of NW Africa is influenced by increased winter rainfall contemporaneous to an intensified monsoon during insolation maxima with a possibly overlapping zone of 2 rainfall regimes in the Sahara (Cheddadi et al., 2021, PNAS). With 2 interfering moisture sources carrying different isotope compositions any conversion of dD into rainfall amounts as done in this manuscript (and other studies before) cannot be done. Moisture sourced from the mid-latitude Atlantic likely explains the obliquity component in the hydroclimatic forcing.

As discussed in the added provenance section of the methods, the $\delta\text{D}_{\text{precip}}$ signature in North Africa during dry intervals is similar to our reconstructed $\delta\text{D}_{\text{precip}}$ and $\sim 10\text{‰}$ more enriched during humid intervals meaning our record is likely a conservative estimate of monsoon rainfall variability south of the Sahara. We do recognize the challenge of converting from $\delta\text{D}_{\text{precip}}$ to rainfall estimates from a Holocene-based calibration, and highlight this uncertainty more prominently in the text. For the purpose of this study, the conversion to rainfall is done only to highlight the stark difference in orbital scale variability in precipitation and those estimates for the coming century, which are nearly an order of magnitude different. So even with large uncertainties we believe that it is still instructive to make this general comparison.

The idea of insolation gradients as drivers of hydroclimate changes in NW Africa is presented and extensively discussed in Kuechler et al. (2018, Climate of the Past). While this manuscript now comes to slightly different conclusions than Kuechler et al. (2018) (interhemispheric tropical versus latitudinal insolation gradient), it reads as if the idea presented here is new. If the slightly different findings to Kuechler et al. (2018) warrants a dedicated publication they should be discussed in more detail in a more specialized journal.

In the main text, we more explicitly describe the findings of the Kuechler et al. (2018) paper and provide evidence from climate model simulations showing that increased cross-equatorial moisture transport and not reduced moisture transport out of the subtropics more likely explains the obliquity response we observe in the northwest African monsoon rainfall.

Dust fluxes:

Aeolian dust needs deflatable source areas AND strong winds to result in elevated dust fluxes to offshore areas. If either of the components is not given, no elevated dust fluxes will be created. As extensively discussed in Trauth et al. (2009, QSR) dust therefore mainly is an ‘aridification

proxy' indicating deflation of previously vegetated areas under strong wind conditions. It is not a measure of aridity itself as under prolonged drought conditions dust source areas will be depleted. The authors present new estimates of dust fluxes based on proxy-normalization which show a strong anti-correlation with inferred rainfall intensity contradicting the earlier view of dust generation and transport.

The manuscript states: "Contraction of vegetated areas and reduced soil moisture during weak monsoon intervals (see below) expands the total area of deflatable sediment, increasing the potential for higher dust emission. At the same time, stronger winds during dry intervals⁴⁶ likely also contribute to higher dust fluxes when δD_{precip} indicates decreased rainfall." This captures the dust production paradigm described by Trauth et al. (2009).

No discussion is made about the significance of the new dust proxy. Findings in this manuscript are repeated from Skonieczny et al. (2019, Science Advances) presenting data from the same sediment core. I cannot see how this part of the manuscript is adding new insights nor is connected to the prior or following part of discussion.

The conclusions of the Skonieczny et al. (2019) study are similar to these presented here in that they both provide further evidence that constant flux normalization is critical to reconstructing dust fluxes in this region. Age-model derived dust fluxes result in biased fluxes on glacial/interglacial time scales due to syndepositional processes such as sediment winnowing and focusing. However, while Skonieczny et al. (2019) similarly conclude that dust fluxes off Northwest Africa do not have a primarily glacial/interglacial frequency, they attribute changes in these fluxes to changes in high latitude Northern Hemisphere insolation. This manuscript shows that changes in the subtropical energy budget driven by local insolation and the summer cross equatorial insolation gradient are the main control on both the dust flux and monsoon rainfall. Further we show that the data from Skoneiczny et al. (2019) are in fact better explained by this mechanism than by changes in high latitude northern hemisphere summer insolation.

C3/C4 occurrence related to pCO₂:

Since the Pliocene the broad distinction of African photosynthetic vegetation types are C3 trees and C4 (warm-season) grasses. C3 (cold-season) grasses only occur in specific areas such as the Mediterranean coast and the western Cape receiving substantial amounts of winter rainfall. The occurrence of trees and shrubs (C3) is limited by the length of the dry season. A long dry season will not allow trees and shrubs to survive. That is why the length of the dry season is the primary control of the occurrence of woody vegetation (C3) in modern African savannahs (see supplement to Collins et al., 2011, Nature Geoscience). With no large pCO₂ change from the mid- to Late Holocene (excluding modern data) a large C3 increase is detected in savannahs of both Hemispheres (Collins et al., 2011) testifying that this principle also holds true in the past. For the LGM, a C4 increase is detected in savannahs on both hemispheres (Collins et al., 2011) correlating to vast evidence for overall drier conditions during the LGM (e.g., Gasse et al., 2000, QSR).

The compilation in this paper (see fig 4) for latitudes north of 10°N is consistent with that of Collins et al. (2011) and the Kuechler et al. (2013) interpretation of decreased area of grasslands as the Sahara Desert expanded at the end of the African humid period. The manuscript cites these papers and describes how changes in rainfall drove this relative C₃ increase from middle to late Holocene when atmospheric CO₂ concentration did not appreciably change.

The increase in C₄ during the LGM at similar latitudes can be explained by decreased CO₂ resulting in lower growth rates for C₃ trees and subsequent increased competitive advantage for C₄ grasses and does not require a change in aridity, thus using an increase in C₄ as evidence of increased aridity is fraught.

Gasse et al. (2000) includes only one terrestrial record of rainfall variability from Northwest Africa: a lacustrine record of bulk organic $\delta^{13}\text{C}$ from Lake Bosumtwi in Ghana (Talbot & Johannessen, 1992). Gasse et al. (2000) interpreted increased bulk organic $\delta^{13}\text{C}$ reconstructed in this study during the LGM as representing more arid conditions in the region. However, these more positive $\delta^{13}\text{C}$ values, representing more grassy C₄ plants in the region, could also be explained by lower atmospheric CO₂ concentrations during this interval favoring lower woody cover in savanna ecosystems. Moreover additional reconstructions of Niger river discharge sampling rainfall in a broad region of Northwest Africa from the LGM to the late-Holocene show no difference between glacial and late-Holocene river discharge (Pastouret et al., 1978). Recent compilation of African lake levels from the Oxford lake level database over the past 20 kyrs do not show significantly lower lake levels during the LGM compared to the late Holocene, see fig. 2 in (deMenocal, P. B. & Tierney, J. E. (2012) Green Sahara: African Humid Periods Paced by Earth's Orbital Changes. *Nature Education Knowledge* 3(10):12). The evidence for the similarity of LGM and recent precipitation in West Africa is outlined and cited in the manuscript.

Dry season length does impose restrictions, however competitive interactions are thought to be the driver of the woody cover/rainfall relationship in bi-stable savannas (Staver et al., 2011). CO₂ is also recognized as a contributor to these interactions based on its impact on growth rate and photosynthetic yields which can alter the relationship between rainfall and C₃ tree v. C₄ grass cover in savanna ecosystems (Bond and Midgley, 2000; Scheiter et al., 2012; Bellasio et al. 2018; etc.). These multiple factors are described in the text.

The authors now propose (based on the quantum yield model from Ehleringer et al., 1997, *Oecologia*) that “the physiological control of atmospheric pCO₂ on photosynthesis dominantly controls the woody/grassy balance in existing savannahs”. This is a re-iteration of arguments proposed by Bragg et al. (2013, *Biogeosciences*) who used the same re-interpretation of data originally presented by Collins et al. (2011).

References to Bragg et al. (2013) have been added to the manuscript. Bragg et al. use isotope guided vegetation models to show that the southwest African portion of the Collins et al. (2011) $\delta^{13}\text{C}$ transect data cannot be explained by rainfall and temperature change alone and CO₂-forced changes in plant physiology are required to simulate the observed changes in the vegetation reconstruction. In this study, we propose similar arguments as Bragg et al. (2013),

but show that the CO₂-forcing mechanism of savanna woody cover holds both in the Northwest African monsoon region and for time periods prior to the LGM, likely for at least the last ~500 ka.

Obviously, the pCO₂ effect on C₃/C₄ would be an alternative explanation of the C₄ increase during the LGM but only if total rainfall (or wet season length) remained the same. Ascribing the entire C₄ increase during the LGM to a pCO₂ effect as done by the authors (Fig. 4) is highly unrealistic in the light of independent data signaling increased aridity. Why is there a C₃ increase at the LGM in the equatorial rainforest? This speaks against a control by pCO₂. The authors' statement cited above is thus not true.

Please see our above comment, which shows why we do not believe there is vast evidence of a drier LGM in Western Africa. With respect to the increased C₃ signature adjacent to the Congo Rainforest, please see added language in the "C₃/C₄ vegetation dynamics reflect combined influence of pCO₂ and rainfall" section of the manuscript for a possible explanation. But in short, we argue that the overall pattern of nearly continent-wide increase in C₄ vegetation provides compelling evidence of a CO₂ forcing of ecosystem structure.

Obviously, this consideration does not rule out that there may be an effect of 'woody thickening' in tree savannahs due to CO₂ fertilization under otherwise relatively stable hydroclimatic conditions. This effect, however, is difficult to disentangle in modern times from land-use changes or changes in agricultural/farming practice. The data from the past (with a dominant control of wet season length on C₃/C₄) do not help to forecast pCO₂ effects.

This comment is well taken and we have restructured this section and removed explicit predictions of woody cover from the discussion.

Other comments:

I wonder how the authors disentangle pCO₂ changes from changes in global ice volume (which are tightly coupled to each other, e.g. Petit et al., 1999, Nature). Increases in global ice volume act to symmetrically increase the latitudinal temperature gradient, compressing the Hadley circulation which restricts the latitudinal migration of the rainbelt and thus lead to decreased wet season length in both hemispheres.

Please see our comment above to Reviewer #1. In brief, we have added ED figure 7 showing the observed changes in our δD_{precip} monsoon rainfall record lead changes in ice volume by several kyrs, precluding ice sheet variability as a control on the monsoon response and implicating a direct control from low-latitude insolation.

The authors' argument that "During the LGM...rainfall as estimated by plant-wax dD was broadly similar to modern" ignores decades of work by other disciplines (e.g., Gasse, 2000) showing a generally dry LGM in Africa. Besides, it is a mis-understanding as dD of leaf waxes reflects rainfall intensity during the growing season and tells nothing about mean annual

rainfall amount. The dD data corresponding to the time-slice $\delta^{13}\text{C}$ data (Collins et al., 2013, QSR) show a southward shift in rainfall intensity (but not rainfall amounts) during the LGM which is de-coupled from the change in wet season length.

Please see our above comment, which shows that there is not vast evidence of a drier LGM in the region covered by our study. As addressed in a previous comment, while dry season length does impose restrictions on woody C_3 growth, competitive interactions (in part controlled by atmospheric CO_2 concentration) are thought to be the driver of the woody cover/rainfall relationship in bi-stable savannas (Staver et al., 2011)—the main ecosystem type in our study region, so we are cautious to interpret $\delta^{13}\text{C}_{\text{wax}}$ as reflective only of wet season length given the evidence of the independent role of CO_2 on ecosystem C_3/C_4 balance. Moreover, we do not claim to reconstruct mean annual rainfall and are careful throughout the text we have made sure in this revised version to only refer to our interpretation of the $\delta\text{D}_{\text{precip}}$ as “monsoon”, “rainy season”, or “wet season” rainfall, in which case rainfall amount and intensity are synonymous.

The ‘surprising’ finding of C_4 grass expansion into the Sahara Desert during wet phases is already described and explained by Kuechler et al. (2013).

We have now made it clear in the main text that this idea was described in Kuechler et al. (2013). We have changed the language from “surprising” to “counterintuitive” and state that we agree with and support Kuechler et al. (2013) that this is the main driver of the C_4 change associated with monsoon strength variability.

The quantitative conversion of dD_{wax} to $\text{dD}_{\text{precip}}$ as proposed by Niedermeyer et al (2016, GCA) which is used in this manuscript is valid only for the Holocene and large uncertainties arise for other times due to the stability of the amount effect under different boundary conditions, other interfering moisture sources, etc. This is clearly noted in Niedermeyer et al. (2016).

As described above, we recognize the large uncertainty associated with any attempt to convert $\delta\text{D}_{\text{precip}}$ to actual estimates of rainfall. However, for the reasons described previously we believe this is a conservative estimate of the rainfall change and the purpose of making this conversion in this analysis is to broadly compare to future estimates of rainfall variability in this region, which are about an order of magnitude lower than those calculated to occur on orbital timescales in this study.

Reviewers' Comments:

Reviewer #1:

Remarks to the Author:

Please excuse my lateness in submitting this review, the due date came during AGU, and then I spent the holidays with COVID-19.

O'Mara et al. have done a great job of addressing the suggestions from the first round of review. In particular, the discussion of insolation drivers is a lot more extensive and complete. I also find the revised discussion of future changes in vegetation much more balanced and reflective of the uncertainty we have w/r/t to greening and also the role of human modification of the landscape.

I only have a few small comments:

1(Line 30-31: This sentence seems like a remnant from the previous draft and doesn't reflect the nuance that is now written into the paper (which is well-balanced now). Please update this to reflect the final portion of the discussion better.

Figure 1: r^2 values are still listed in the figure caption. Should be r-values to match the main text.

Looking forward to seeing this work published.

-Jess Tierney

Reviewer #2:

Remarks to the Author:

Please see uploaded pdf.

Review of 'Past and future drivers of Northwest African hydroclimate and vegetation' by O'Mara et al.

This is my second review of this manuscript for which I recommended rejection based on unsupported claims. Now the authors provide a revised version which has been re-worded but essentially contains the same arguments and conclusions. I jump directly to the main point.

The authors state that "During the LGM ... rainfall ... was broadly similar to modern". As outlined in my first review and referring to the strong relation between C3/C4 occurrence on the African continent and wet-season length in modern observational data (as analyzed in the supplement of Collins et al., 2011, Nature Geo) a pCO₂ control of C3/C4 distribution can only be inferred under conditions when rainfall remained unchanged, i.e. for this case if the LGM indeed was as dry as the late Holocene. The authors claim that this is the case. But is this true? It was my fault to cite Gasse et al., 2000 in my first review which made the authors discuss a single record in their response. Obviously, I meant the large compilation of data, mainly lake levels which are the best independent evidence for changes in rainfall (or P-E) (in contrast to isotopic changes and/or runoff changes which all can be influenced by several other factors) by Françoise Gasse in 'Hydrological changes in the African tropics since the Last Glacial Maximum' (QSR, 2000). One conclusion from this large review is "... much of the continent experienced cooler conditions than today during glacial times, and a marked decrease in P or in P-E at the LGM". Can such strong statement just be ignored? If 'the LGM was as dry as today' is the 'hidden message' of this manuscript the authors should discuss more openly and in more detail why they discard decades of prior studies and findings. Later follow-up papers scrutinize this finding in more detail. Below a plot from Gasse et al. (QSR, 2008) showing red dots at sites where the LGM was drier than present for the equatorial and southern African continent:

Except for the wetter winter rainfall zone in South Africa I see evidence for drier conditions at almost all sites.

The same is true for Northwest Africa. There are almost no records extending into the LGM for Northwest Africa simply because the area was extremely dry then. Please see below for a plot from Lezine et al. (2011, QSR) showing that NW Africa was certainly drier than today for the LGM.

While I appreciate that pCO₂ can have an influence on C3/C4 in savannahs under stable rainfall conditions as indicated in my first review the re-interpretation of the Collins et al. (2011, Nat Geo) data cannot be used for the purpose of showing a dominant pCO₂ control of the woody/grass balance in savannahs as done by the authors and claimed earlier by Bragg et al., (2013, Biogeosciences) simply because there is evidence that the LGM was drier than today. A drier LGM makes sense due to the higher latitudinal temperature gradients compressing atmospheric circulation and biomes towards the equator and restriction of the seasonal latitudinal migration of the ITCZ during that time. Thus, if the LGM was drier, as ample evidence shows, then there is reason to assume that rainfall amount and seasonality will affect C3/C4 occurrence, as observed today. Disentangling hydrological from pCO₂ effects is then not possible. I appreciate the discussion of pCO₂ effects on climate, vegetation, etc. under current climate change but presenting these data in a biased way as done by the authors provides disservice to the discussion.

Similarly to my first review I do not see how the other aspects discussed in this manuscript are connected to the above discussed main point. ODP659 and MD02-2705 are both located offshore the modern Saharan/Sahelian boundary and are thus influenced by different processes than stable savannah ecosystems. The high dust fluxes in these cores document the deflation of vegetation-barren areas during times of aridification while C4 expansion testifies more northerly monsoonal rainfall during wetter times. I cannot see how these two observations, which already have been described before, are related to the main point of a pCO₂ effect on C3/C4 occurrence in this manuscript. Some of these aspects are interesting, for instance which gradient/s or insolation controls hydrology and dust fluxes, but they are not connected to the main point and demand a more extensive discussion not allowed by the short format of Nature Comm.

In summary, I see no reason why Nature Comm should publish unsupported claims which have already been published before mixed with side aspects which are not adequately discussed and thus recommend rejection.

Reviewer #3:

Remarks to the Author:

O'Mara et al. provide new data from a marine sediment core off northwestern Africa that displays precessional and orbital shifts in NW Africa aridity based on leaf wax δD and normalized dust fluxes. The orbital response is convincingly attributed to the cross-hemispheric insolation gradient and its impact on moisture transport to NW Africa. Leaf wax $\delta^{13}C$, a measure of the relative abundance of C3 woody vegetation and C4 grasses, is shown to correlate strongly with atmospheric CO₂, suggesting that this and not aridity has dominated the prevailing vegetation in NW Africa through time.

I read the comprehensive reviews by J. Tierney and another reviewer of an earlier draft of this paper, along with the extensive changes that were made in response to those reviews. I believe that the authors adequately address the two reviewers' concerns, so I have very little to add to what has already been adopted.

The strength of this paper lies in the excellent sets of δD , dust, and $\delta^{13}C$ data that were generated, and the convincing arguments for linking the first two data sets to the cross-hemispheric insolation gradient and the third data set to the history of atmospheric CO₂ over the time interval of study. The age model for this core is robust and the relative timing of aridity in NW Africa vis a vis NH ice sheet volume (ED Fig. 7) is an important observation. For these reasons alone I support publication in Nature.

The weak aspect of this paper is the attempted forecast of C3 vegetation dominance over C4 vegetation in the Sahel as atmospheric CO₂ continues to rise. While CO_{2atm} clearly influenced the relative abundances of C3 and C4 vegetation in pre-industrial, glacial-interglacial times, we are now in an era of much higher and rising CO₂ concentrations, when soil moisture availability may dominate the prevalence of grass or woody vegetation. There is already a diverse scientific literature that provides reasons why woody vegetation or grasses will prevail in the Sahel in the coming decades, including rising CO₂ and the impact of lush grass inhibiting woody vegetation seedling growth, or woody overstory shading out grass, or changing hydroclimate from "wet Sahel" to "dry Sahel", or over grazing and other land-use practices, and more. This paper by O'Mara et al. offers nothing new to the vegetation forecast and therefore should delete "And Future" from the title and should keep any prediction of future vegetation change to a minor paragraph in the paper.

Some minor comments:

Lines 28-31: As J. Tierney pointed out, pCO₂ is now well above concentrations that clearly impact the relative abundances of C3 and C4 plants. Delete this sentence.

Line 190, references to Figs 3: I don't see shading in Figs. 3 a and c that is mentioned in the figure caption. It would be nice to see the 23.5°N insolation curve along with the insolation gradient curve, as you have portrayed in Fig. 1.

Line 219: You have only one data point poleward of 15°S –you should delete "/S" from this sentence.

Lines 281-285: As in lines 28-31, does pCO₂ have any influence on the relative viability of C3 vs C4 plants now that we are well above 400 ppm? Rainfall and soil moisture may now be the dominating influence on the relative abundance of C4 and C3 vegetation on the landscape.

Line 724: Specifically, what values were used for $\delta^{13}C_{C4}$ and $\delta^{13}C_{C3}$?

Line 732: How do wC4 and fC4 relate to one another? Are they the same value? If so, then just use fC4 in the equation. If not, explicitly state the relationship. Ditto, of course, for wC3 and fC3.

REVIEWER COMMENTS

Reviewer #1 (Remarks to the Author):

Please excuse my lateness in submitting this review, the due date came during AGU, and then I spent the holidays with COVID-19.

O'Mara et al. have done a great job of addressing the suggestions from the first round of review. In particular, the discussion of insolation drivers is a lot more extensive and complete. I also find the revised discussion of future changes in vegetation much more balanced and reflective of the uncertainty we have w/r/t to greening and also the role of human modification of the landscape.

I only have a few small comments:

1) Line 30-31: This sentence seems like a remnant from the previous draft and doesn't reflect the nuance that is now written into the paper (which is well-balanced now). Please update this to reflect the final portion of the discussion better.

Reviewer #3 suggested we simply remove this sentence from the abstract, which we have done.

Figure 1: r^2 values are still listed in the figure caption. Should be r-values to match the main text.

Done, the figure caption has been updated.

Looking forward to seeing this work published.

Thank you so much for your helpful critiques of the manuscript they have led to substantial improvements.

-Jess Tierney

Reviewer #2 (Remarks to the Author):

This is my second review of this manuscript for which I recommended rejection based on unsupported claims. Now the authors provide a revised version which has been re-worded but essentially contains the same arguments and conclusions. I jump directly to the main point.

The authors state that "During the LGM ... rainfall ... was broadly similar to modern". As outlined in my first review and referring to the strong relation between C3/C4 occurrence on the African continent and wet-season length in modern observational data (as analyzed in the supplement of Collins et al., 2011, Nature Geo) a pCO₂ control of C3/C4 distribution can only be

inferred under conditions when rainfall remained unchanged, i.e. for this case if the LGM indeed was as dry as the late Holocene. The authors claim that this is the case. But is this true? It was my fault to cite Gasse **et al.**, 2000 in my first review which made the authors discuss a single record in their response. Obviously, I meant the large compilation of data, mainly lake levels which are the best independent evidence for changes in rainfall (or P-E) (in contrast to isotopic changes and/or runoff changes which all can be influenced by several other factors) by Francoise Gasse in 'Hydrological changes in the African tropics since the Last Glacial Maximum' (QSR, 2000). One conclusion from this large review is "... much of the continent experienced cooler conditions than today during glacial times, and a marked decrease in P or in P-E at the LGM". Can such strong statement just be ignored? If 'the LGM was as dry as today' is the 'hidden message' of this manuscript the authors should discuss more openly and in more detail why they discard decades of prior studies and findings. Later follow-up papers scrutinize this finding in more detail. Below a plot from Gasse et al. (QSR, 2008) showing red dots at sites where the LGM was drier than present for the equatorial and southern African continent:

Except for the wetter winter rainfall zone in South Africa I see evidence for drier conditions at almost all sites.

The same is true for Northwest Africa. There are almost no records extending into the LGM for Northwest Africa simply because the area was extremely dry then. Please see below for a plot from Lézine et al. (2011, QSR) showing that NW Africa was certainly drier than today for the LGM.

Careful analysis of the records presented in the Gasse et al. (2008) review paper for "the West African monsoon domain and southwestern tropics" (defined as region 1 in the paper) reveals that the overwhelming majority of the records rely on the presence/absence of forest/savanna taxa as an indicator of humid/arid conditions during the LGM. However, as noted by Gasse et al. (2008), and a number of recent papers (e.g. Bragg et al. (2013), Prentice et al, (GCB, 2003; GPC, 2017; GPC, 2022)) note that using biome classifications as estimates of past rainfall without considering the effects of low CO₂ levels during the LGM lead to overestimations of aridity. This fact appears to be the case, when lake levels or other direct proxies of aridity are compared with other vegetation-based estimates. We have added a new section to the methods that explains this (see 'Assessing evidence of hydrologic change during the Last Glacial Maximum'). We have also added a supplemental figure (fig. ED8) which clearly demonstrates that our $\delta^{13}\text{C}_{\text{wax}}$ record tracks CO₂, and not ice volume, suggesting that any pacing of rainfall variability that may be set by glacial-interglacial cycles is not the main driver of the vegetation response we reconstruct here.

We agree that lake level records are often the most direct means of estimating changes in past P-E. The cited study from Lézine et al. (2011) however only covers the last 18 kyrs and is primarily focused on changes in lake levels over the course of the deglaciation and Holocene and not focused on the LGM itself. Following Collins et al. (2011) we define the LGM in our study as the interval spanning 23-19 ka, which is not covered by the Lézine et al. (2011) analysis. We refer the reviewer to our previous response where we cite the compilation of

African lake levels published by de Menocal and Tierney (2012) (below) which, although not spanning the full LGM range of 23-19 ka, extends farther back in time to 20 ka and shows that lake levels prior to 18 ka were similar if not higher compared to the late Holocene (2-0 ka). To emphasize that this is true for North Africa specifically I have replotted the data from de Menocal and Tierney (2012) only showing lake level estimates north of 10°N (see locations in bottom panel).

While I appreciate that pCO₂ can have an influence on C3/C4 in savannahs under stable rainfall conditions as indicated in my first review the re-interpretation of the Collins et al. (2011, Nat Geo) data cannot be used for the purpose of showing a dominant pCO₂ control of the woody/grass balance in savannahs as done by the authors and claimed earlier by Bragg et al., (2013, Biogeosciences) simply because there is evidence that the LGM was drier than today. A drier LGM makes sense due to the higher latitudinal temperature gradients compressing atmospheric circulation and biomes towards the equator and restriction of the seasonal latitudinal migration of the ITCZ during that time. Thus, if the LGM was drier, as ample evidence shows, then there is reason to assume that rainfall amount and seasonality will affect C3/C4 occurrence, as observed today. Disentangling hydrological from pCO₂ effects is then not

possible. I appreciate the discussion of pCO₂ effects on climate, vegetation, etc. under current climate change but presenting these data in a biased way as done by the authors provides disservice to the discussion.

Bragg et al. (2013) use a biogeographic model to explicitly test if the more positive carbon isotope values of Collins et al. (2011) observed during the LGM can be explained by rainfall and temperature declines alone. They conclude it cannot and state in their abstract: "Using a process-based biogeography model that explicitly simulates ¹³C discrimination, it is shown that precipitation and temperature changes cannot explain the observed shift in δ¹³C values. The physiological effect of increasing CO₂ concentration is decisive, altering the C₃/C₄ balance and bringing the simulated and observed δ¹³C values into line." This statement is again supported by the hydrologic evidence for the LGM as discussed above.

Similarly to my first review I do not see how the other aspects discussed in this manuscript are connected to the above discussed main point. ODP659 and MD02-2705 are both located offshore the modern Saharan/Sahelian boundary and are thus influenced by different processes than stable savannah ecosystems. The high dust fluxes in these cores document the deflation of vegetation- barren areas during times of aridification while C4 expansion testifies more northerly monsoonal rainfall during wetter times. I cannot see how these two observations, which already have been described before, are related to the main point of a pCO₂ effect on C₃/C₄ occurrence in this manuscript. Some of these aspects are interesting, for instance which gradient/s or insolation controls hydrology and dust fluxes, but they are not connected to the main point and demand a more extensive discussion not allowed by the short format of Nature Comm.

In summary, I see no reason why Nature Comm should publish unsupported claims which have already been published before mixed with side aspects which are not adequately discussed and thus recommend rejection.

As noted above, our claims are supported by the existing literature, and our findings add to the previous work showing the role of both rainfall and CO₂ concentrations on African ecosystems.

Reviewer #3 (Remarks to the Author):

O'Mara et al. provide new data from a marine sediment core off northwestern Africa that displays precessional and orbital shifts in NW Africa aridity based on leaf wax δD and normalized dust fluxes. The orbital response is convincingly attributed to the cross-hemispheric insolation gradient and its impact on moisture transport to NW Africa. Leaf wax ¹³C, a measure of the relative abundance of C₃ woody vegetation and C₄ grasses, is shown to correlate strongly with atmospheric CO₂, suggesting that this and not aridity has dominated the prevailing vegetation in NW Africa through time.

I read the comprehensive reviews by J. Tierney and another reviewer of an earlier draft of this

paper, along with the extensive changes that were made in response to those reviews. I believe that the authors adequately address the two reviewers' concerns, so I have very little to add to what has already been adopted.

The strength of this paper lies in the excellent sets of $\delta^{18}O$, dust, and $\delta^{13}C$ data that were generated, and the convincing arguments for linking the first two data sets to the cross-hemispheric insolation gradient and the third data set to the history of atmospheric CO_2 over the time interval of study. The age model for this core is robust and the relative timing of aridity in NW Africa vis a vis NH ice sheet volume (ED Fig. 7) is an important observation. For these reasons alone I support publication in Nature.

We very much appreciate your positive review. Thank you for your helpful critiques of our future implications section. We believe our now more nuanced framing of the role of CO_2 on future savanna woody cover, taken at your suggestion, has improved the quality of the manuscript.

The weak aspect of this paper is the attempted forecast of C3 vegetation dominance over C4 vegetation in the Sahel as atmospheric CO_2 continues to rise. While CO_2 clearly influenced the relative abundances of C3 and C4 vegetation in pre-industrial, glacial-interglacial times, we are now in an era of much higher and rising CO_2 concentrations, when soil moisture availability may dominate the prevalence of grass or woody vegetation. There is already a diverse scientific literature that provides reasons why woody vegetation or grasses will prevail in the Sahel in the coming decades, including rising CO_2 and the impact of lush grass inhibiting woody vegetation seedling growth, or woody overstory shading out grass, or changing hydroclimate from "wet Sahel" to "dry Sahel", or over grazing and other land-use practices, and more. This paper by O'Mara et al. offers nothing new to the vegetation forecast and therefore should delete "And Future" from the title and should keep any prediction of future vegetation change to a minor paragraph in the paper.

The title was altered from "Past and future" to "Pleistocene" to better reflect the toned-down section on forecasting future savanna woody cover.

In accordance with this reviewer's suggestion, the final section of our paper "Implications for the future of Northwest African ecosystems" has been reduced to a single paragraph. We have also noted that the CO_2 fertilization effect likely saturates at values of ~400-500 ppm, making the driving role of CO_2 on future increases of woody encroachment uncertain. We have completely removed the section projecting a continued rise in savanna woody cover with CO_2 . We have replaced these projections with language indicating instead that persistently high atmospheric CO_2 levels will serve as a continued ecological pressure on current savanna woody cover.

Some minor comments:

Lines 28-31: As J. Tierney pointed out, pCO₂ is now well above concentrations that clearly impact the relative abundances of C₃ and C₄ plants. Delete this sentence.

This sentence was deleted from the abstract.

Line 190, references to Figs 3: I don't see shading in Figs. 3 a and c that is mentioned in the figure caption. It would be nice to see the 23.5°N insolation curve along with the insolation gradient curve, as you have portrayed in Fig. 1.

Figures 3a and 3c do in fact have the shading mentioned in the figure caption. The uncertainty for the Kuechler et al. (2013) δD data was calculated based only on replicate precision of external standards and the C₃₁ *n*-alkane homolog, while here we take the more conservative approach of Polissar and D'Andrea (*GcA*, 2014) and calculate error bars by propagating uncertainties not only in laboratory precision but also in the reference gas and the VSMOW standard, resulting in larger error bars that more accurately reflect the absolute δD value on the VSMOW scale. In figure 3c, again the shading is there, the error in ²³⁰Th_{X_S}-normalization is just much lower than ³He_{ET} resulting in much smaller error bars comparatively.

The local insolation curve has been added to figure 3.

Line 219: You have only one data point poleward of 15°S –you should delete “/S” from this sentence.

“/S” and reference to the Namib Desert were removed from this sentence.

Lines 281-285: As in lines 28-31, does pCO₂ have any influence on the relative viability of C₃ vs C₄ plants now that we are well above 400 ppm? Rainfall and soil moisture may now be the dominating influence on the relative abundance of C₄ and C₃ vegetation on the landscape.

Future projections of a continued rise in savanna woody cover have been deleted and instead we take the more conservative stance the high CO₂ levels will serve as an ecological pressure favoring high woody covers. See above response and restructured “Implications for the future of Northwest African ecosystems” section.

Line 724: Specifically, what values were used for $\delta^{13}C_{C4}$ and $\delta^{13}C_{C3}$?

End member values for the C₄ and C₃ end members were added to this sentence. $\delta^{13}C_{C4} = -19.9\text{‰}$ and $\delta^{13}C_{C3} = -33.6\text{‰}$.

Line 732: How do wC₄ and fC₄ relate to one another? Are they the same value? If so, then just use fC₄ in the equation. If not, explicitly state the relationship. Ditto, of course, for wC₃ and fC₃.

You are correct, these are the same variables. Thank you for pointing this out. For consistency wC3 and wC4 were changed to fC3 and fC4.

Reviewers' Comments:

Reviewer #3:

Remarks to the Author:

I have reviewed the revised manuscript and find that the authors adequately addressed the concerns that I raised in my review of the earlier draft. I agree with Jess Tierney that this revision should be published in Nature. I believe that the authors adequately address the concerns initially raised by Reviewer #2, and I do not agree with Reviewer #2's conclusion that the C3/C4 vegetation history was influenced more by P-E than by CO₂atm. Gasse et al. (2008) presents data from lakes that extend no farther north in Africa than 6.5° N. This paper centers on ~15°N. The δD and dust records track one another closely, and are probably good indicators of the rainfall history in the nearby NW African sahel, and clearly do not track $\delta^{13}C$, which does track CO₂atm.

REVIEWER COMMENTS

Reviewer #3 (Remarks to the Author):

I have reviewed the revised manuscript and find that the authors adequately addressed the concerns that I raised in my review of the earlier draft. I agree with Jess Tierney that this revision should be published in Nature. I believe that the authors adequately address the concerns initially raised by Reviewer #2, and I do not agree with Reviewer #2's conclusion that the C3/C4 vegetation history was influenced more by P-E than by CO₂atm. Gasse et al. (2008) presents data from lakes that extend no farther north in Africa than 6.5° N. This paper centers on ~15°N. The δD and dust records track one another closely, and are probably good indicators of the rainfall history in the nearby NW African sahel, and clearly do not track $\delta^{13}C$, which does track CO₂atm.

Author response:

We thank the reviewer for the positive assessment of our manuscript and the helpful comments provided in the previous round of reviews.